# EvolvedGRPO: Unlocking Reasoning in LVLMs via Progressive Instruction Evolution

**Zhebei Shen**[1][*]   **Qifan Yu**[1][*]   **Juncheng Li**[1][†]   **Wei Ji**[2]   **Qizhi Chen**[3]
**Siliang Tang**[1]   **Yueting Zhuang**[1]

[1]Zhejiang University   [2]Nanjing University   [3]Peking University
{shenzhebei, yuqifan, junchengli, siliang}@zju.edu.cn

## Abstract

Recent advances in reinforcement learning (RL) methods such as Grouped Relative Policy Optimization (GRPO) have strengthened the reasoning capabilities of Large Vision-Language Models (LVLMs). However, due to the inherent entanglement between visual and textual modalities, applying GRPO to LVLMs often leads to reward convergence across different responses to the same sample as training progresses, hindering effective gradient updates and causing the enhancement of chain-of-thought reasoning to stagnate or even collapse. To address this issue, we propose a progressive instruction evolution framework, EvolvedGRPO, to gradually generate more complex questions via editing instructions in an adversarial way, progressively aligned with the model's evolving capabilities. Specifically, we design two instruction editing strategies across modalities, incorporating incrementally increasing editing instructions and RL-based adversarial data augmentation to improve the effectiveness of model training. To address GRPO's limitations on overly difficult problems, we first train on basic subproblem versions of complex multi-modal questions in both the visual and textual modalities, progressively increasing difficulty to enable prefix-style process rewards, effectively combining the strengths of both process rewards and group-wise relative rewards. Finally, EvolvedGRPO achieves state-of-the-art performance among open-source RL models on multi-modal reasoning tasks, even approaching the closed-source GPT-4o in reasoning capabilities, and demonstrates better performance on unseen LVLM general benchmarks. The Code for EvolvedGRPO is available at https://github.com/SHENZHEBEI/EvolvedGRPO.

## 1   INTRODUCTION

Large Language Models (LLMs) have achieved remarkable progress in multi-step reasoning tasks, most notably through models such as OpenAI's GPT-4o [1]. Building on this progress, recent work DeepSeek-R1 [2] highlights that reinforcement learning (RL) with verifiable rewards is particularly effective in eliciting nuanced self-verification and self-correction behaviors in LLMs, substantially enhancing the reliability of mathematical and logical reasoning chains.

Going a step further, with the emergence of advanced large vision-language models (LVLMs), there has been a further RL exploration into LVLMs [3, 4, 5, 6, 7] to enhance their reasoning abilities. However, general RL methods such as Grouped Relative Policy Optimization (GRPO) necessitate assigning varying reward scores to different responses from the same prompt to generate meaningful

---

[*]Equal contribution.
[†]Corresponding author.

39th Conference on Neural Information Processing Systems (NeurIPS 2025).

gradients. Yet in multi-modal tasks, unlike purely textual reasoning tasks that benefit from clear formats, making stepwise reward design feasible, the entanglement between visual and textual modalities poses fundamental challenges for assigning such differentiated rewards.

Specifically, as the LVLM's visual reasoning ability improves over multiple rounds of training, the model gradually memorizes both visual and textual information from the training dataset, enabling it to recall answers solely from the reasoning processes it has internalized during previous training, thereby reducing the need for genuine perception and reasoning. This results in low intra-group reward variance under the GRPO strategy, making effective gradient updates difficult and causing the reasoning capability to stagnate or even collapse [4] As a result, the GRPO strategy in multi-modal tasks struggles to maintain sufficient intra-group reward variance as training progresses, and two problems are notably amplified. (a) For relatively simple samples, different model responses tend to yield uniformly high rewards despite varying reasoning paths, due to the absence of a mechanism to reward deeper or more faithful visual grounding. This leads to premature reward convergence and provides limited learning signal for improving multi-modal reasoning capabilities. (b) For overly complex samples, the model often fails to construct valid long-horizon reasoning chains that jointly interpret visual and textual cues. Since only final outputs are evaluated, all failed attempts may receive equally low rewards, making it difficult for the policy to learn incremental improvements.

To tackle these challenges faced by GRPO in multimodal reasoning, previous works mainly focus on either distilling knowledge from stronger external models [3, 6, 8, 9, 10] or collecting more challenging multi-modal perceptual datasets [11, 12, 13, 14] to adapt to the evolving capabilities of LVLMs. However, such adaptation paradigms either depend on the performance upper bound of external teacher models or incur substantial human labor costs for data construction. Moreover, the inherent disconnect between external teacher models and the internal answering model limits the potential to fully exploit the model's reasoning capabilities, resulting in training data that fails to keep pace with the model's evolving abilities.

In this paper, we propose a progressive instruction evolution framework (**EvolvedGRPO**) that iteratively increases the difficulty of samples through self-evolved editing instructions with low overhead to overcome the challenges of reward saturation and adapt to the model's continually advancing reasoning capabilities. Based on this evolutionary framework, we observe that the model can gradually improve its performance through adversarial interactions within the framework. Formally, we use one model as the question generator $\mathcal{Q}$, which is used to generate two types of multi-modal data editing instructions aimed at augmenting the dataset with learnable multi-step reasoning tasks that remain within the model's capability range: (1) **Image editing instructions** $I_v$, using external tools such as Diffusion Models [15] are employed to edit the training image corresponding to the instruction $I_v$, thereby accurately recognizing and analyzing visual information; (2) **Text editing instructions** $I_t$, which are designed to alter the original question-answer pair, forcing a transformation of the original answer and introducing multi-step reasoning into the resulting multi-modal question, thereby increasing the number of reasoning steps required. Meanwhile, we employ another model as the answer model $\mathcal{A}$, which is trained to solve multi-modal reasoning problems via chain-of-thought reasoning. In each iteration, we generate an instruction pool $\mathcal{I}$ and randomly sample $k$ mixed instructions to synthesize each multi-modal question based on an original question. Benefiting from the Markov property, the trustworthy answer is derived in a step-by-step manner using a TTRL [16] applied to the outputs of the trained model.

Moreover, we further effectively enhance the instruction evolution to activate LVLMs' deeper reasoning ability with two progressive training strategies: (1) **train the two models adversarially through reinforcement learning**, using the accuracy of answer model $\mathcal{A}$ as the critic of question generator $\mathcal{Q}$ to improve the quality of each generated instruction; (2) **gradually increase the number of sampled instructions** $k$ to progressively guide the model toward longer $k+1$-step reasoning chains in a curriculum learning manner. Under such a progressive promotion, extensive experiments on challenging multimodal reasoning benchmarks and general LVLMs benchmarks show that EvolvedGRPO effectively enables the model to explore the most suitable chain-of-thought paths, thereby unlocking continual growth and qualitative improvements in reasoning trajectories. Overall, our main contributions are three-fold:

- We propose **EvolvedGRPO**, a progressive instruction evolution framework leverages a GRPO-trained question generator to apply multi-modal editing instructions, constructing increasingly challenging datasets tailored to the model's evolving reasoning capabilities.

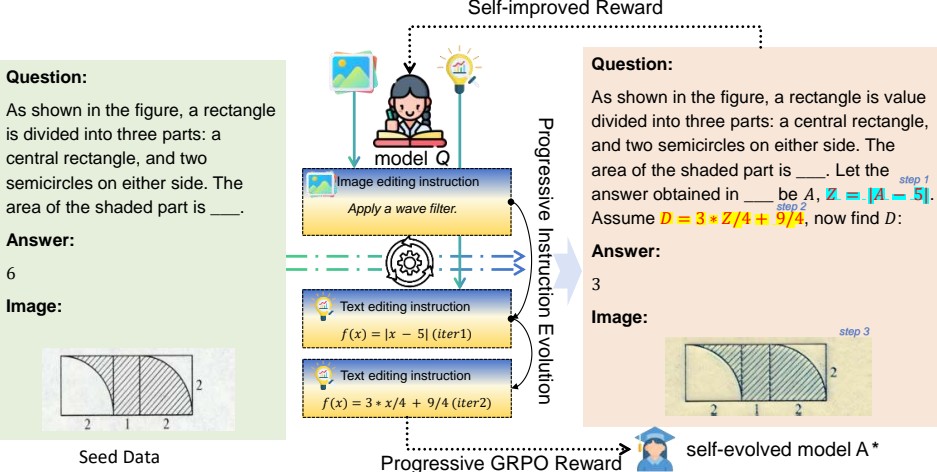

Figure 1: Overview of our proposed EvolvedGRPO for enhancing the model's reasoning capability by progressively using instruction evolution during GRPO-based training in an adversarial way.

- We adversarially train the question generator and answer model with curriculum-style instruction scaling, enabling continuous improvement of LVLMs' reasoning capabilities.
- Our method ensures that, across diverse tasks and reasoning scenarios, LVLMs can consistently unlock their underlying reasoning capabilities in a self-driven manner with minimal overhead.

## 2 RELATED WORK

**Equipping LVLMs with Reasoning Ability.** With the remarkable generalizability of LLMs in a zero-shot setting, Large Vision-Language Models (LVLMs) [17, 18, 19, 20, 21, 22, 23, 24, 25, 26, 27, 28, 29] integrating LLMs with visual modality have demonstrated impressive visual-language understanding ability across diverse scenarios [30, 31, 32, 33]. Motivated by the significant advances in reasoning demonstrated by LLMs, substantial efforts have been directed toward enhancing the reasoning capabilities of LVLMs via supervised fine-tuning (SFT) [34, 35, 36, 37, 38] and chain-of-thought (CoT) prompting [38, 39, 40, 41]. More recently, Reinforcement Learning (RL) approaches such as GRPO [2] have emerged as an effective strategy to unlock underlying reasoning abilities in LVLMs [3, 4, 5, 6, 7]. Despite these advancements, applying RL to multi-modal tasks remains challenging due to the difficulty of defining fine-grained rewards across entangled modalities, which limits the full exploitation of reasoning capabilities in LVLMs. This is essential for establishing such granular reward signals to unlock reasoning ability in LVLMs.

**Multi-model Reasoning Instruction Dataset.** Although reasoning LVLMs have demonstrated remarkable performances in various VL reasoning tasks, they still require efficient utilization of high-quality data to enable robust and transferable reasoning capabilities. Recent methods rely heavily on external powerful models (*e.g.*, GPT-4o, DeepSeek-R1) to synthesize multi-step questions [10] or long chain-of-thought reasoning instructions [3, 6, 8, 9], essentially performing pre-trained knowledge distillation, which consequently limits the upper bound of the model's reasoning ability. Furthermore, several works [11, 12, 13, 42, 43] focus on collecting more challenging multi-modal perceptual datasets via tool-invoking methods [44, 45, 46, 47, 48, 49] for reasoning instruction construction, but require substantial human labor costs for data collection. Generally, these methods suffer from limited generalization across modalities and tasks, failing to establish a tight coupling between visual and textual semantics. In contrast, we propose an instruction evolution method that progressively expands the reasoning capabilities by leveraging an adversarial instruction synthesis strategy based solely on the LVLM itself, naturally adapting to the evolving needs of the model.

# 3 PRELIMINARIES

**Large Vision-Language Models (LVLMs)** LVLMs represent a class of multi-modal large language models that integrate both visual and textual information as input, generating textual outputs. The model architecture of LVLMs commonly consists of three components: large language model, vision encoder and MLP-based vision-language merger [23]. The LVLMs use vision encoder to transform inputs visual information into visual embeddings, and use MLP-based vision-language merger to align and integrate the visual embeddings and textual embeddings into a unified feature space. Finally, the large language model autoregressively generates the output.

**Supervised Fine Tuning (SFT).** As the most commonly used technique for LLMs and multi-modal LLMs, Supervised Fine Tuning (SFT) is a simple and efficient method to align pre-trained models with downstream tasks. Formally, given a multi-modal dataset $\mathcal{D}$ including question $q$, answer $a$ and image $v$, the training loss of SFT is:

$$\mathcal{L}_{\text{SFT}}(\theta) = -\mathbb{E}_{(q,a,v)\sim D}\Big[ \sum_{i=1}^{L}\sum_{c=1}^{C} \mathbb{I}(a_i = c) \log \pi_\theta(c|q,v,a_{<i})\Big], \qquad (1)$$

where $\theta$ is the parameter of model, $L$ is the token length of the answer $a$, $C$ is the number of candidate token classes determined by sampling methods such as top-$p$ or top-$k$, $\mathbb{I}(a_i = c)$ is the indicator function that equals 1 if the condition $a_i = c$ is true and equals 0 otherwise, and $\pi_\theta(c|q,v,a_{<i})$ represents the model's predicted probability of the $i$-th token equals to $c$ given the question $q$, image $v$ and the sequence of preceding tokens $a_{<i}$.

**Group Relative Policy Optimization (GRPO)** As in LLMs, often the last token typically receives a reward score, Group Relative Policy Optimization (GRPO) reduces the memory and computational burden compared to PPO [50], which eliminates the need for an additional value function approximation. GRPO uses the average reward of multiple outputs, sampled in response to the same question as the baseline. For each question $q$, GRPO samples a group of outputs $\{o_1, o_2, \ldots, o_G\}$ from the old policy $\pi_{\theta_{\text{old}}}$ and optimizes the policy model by maximizing the following objective:

$$\hat{A}_{i,t} = r_i = \frac{r_i - \text{mean}(\{r_j\}_{j=1}^{G})}{\text{std}(\{r_j\}_{j=1}^{G}))}, \quad \forall i \in \{1,\ldots,G\} \qquad (2)$$

where $\hat{A}_{i,t}$ is the advantage calculated based on relative rewards of the outputs within each group.

$$\mathcal{J}_{\text{GRPO}}(\theta) = \mathbb{E}\left[q \sim P(Q), \{o_i\}_{i=1}^{G} \sim \pi_{\theta_{\text{old}}}(O|q)\right]$$

$$\frac{1}{G}\sum_{i=1}^{G}\frac{1}{|o_i|}\sum_{t=1}^{|o_i|}\{\min[r^{i,t}(\theta)\hat{A}_{i,t}, \text{clip}\left(r^{i,t}(\theta), 1-\epsilon, 1+\epsilon\right)\hat{A}_{i,t}] - \beta D_{\text{KL}}[\pi_\theta||\pi_{\text{ref}}]\} \qquad (3)$$

where

$$r^{i,t}(\theta) = \frac{\pi_\theta(o_{i,t}|q,o_{i,<t})}{\pi_{\theta_{\text{old}}}(o_{i,t}|q,o_{i,<t})} \text{ and } D_{\text{KL}}[\pi_\theta||\pi_{\text{ref}}] = \frac{\pi_{ref}(o_{i,t}|q,o_{i<t})}{\pi_\theta(o_{i,t}|q,o_{i<t})} - \log\frac{\pi_{ref}(o_{i,t}|q,o_{i<t})}{\pi_\theta(o_{i,t}|q,o_{i<t})} - 1.$$

Here, $\pi_\theta$ and $\pi_{\theta_{\text{old}}}$ represent the current and old policy models, respectively, and $q$ and $o$ are the questions and outputs sampled from the question dataset and the old policy $\pi_{\theta_{\text{old}}}$. The $\epsilon$ is a clipping-related hyperparameter to stabilize training.

This method leverages the comparative nature of reward models, which are typically trained on datasets containing comparisons between outputs for the same question. Instead of adding a KL penalty directly to the reward, GRPO regularizes by adding the KL divergence between the trained policy and the reference policy to the loss function, simplifying the calculation of $\hat{A}_{i,t}$.

# 4 METHOD

In this section, we first introduce a formulation for evolved editing instructions to dynamically provide evolution intents (§4.1). Based on these, we develop a progressive instruction evolution framework within LVLMs to generate increasingly challenging problems that require longer chains of reasoning (§4.2). Finally, we further enhance the instruction evolution progressive training strategies, unlocking the full reasoning in LVLMs(§4.3). The overall framework is shown in Figure 1.

## 4.1 Formulation for Evolved Editing Instructions

During the multi-round GRPO training with fixed data, the increase in accuracy and the training on fixed problems cause the standard deviation to decrease to zero, leading to reward convergence and, ultimately, training stagnation and collapse. To address this, we introduce EvolvedGRPO, which incorporates adversarial data augmentation to introduce additional information during training.

In general, the paradigm for reasoning instruction involves generating multi-step questions in a single step. We innovatively propose a progressive instruction evolution that incrementally refines instructions for both visual and textual modalities. In this section, we will demonstrate that both image editing instructions and text editing instructions are designed for Progressive Instruction Evolution in LVLMs to unlock their reasoning ability.

Given an initial sample $(q_0, a_0, v_0) \in \mathcal{Q} \times \mathcal{A} \times \mathcal{V}$, where $\mathcal{Q}$, $\mathcal{A}$, $\mathcal{V}$ represent the question, answer, and visual space, respectively. Consider a total of $k$ steps in sample edition, the editing process is recursively defined as:

$$(q_0, a_0, v_0) \xrightarrow{\textbf{Instruction } I_1} (q_1, a_1, v_1) \xrightarrow{\textbf{Instruction } I_2} \cdots \xrightarrow{\textbf{Instruction } I_k} (q_k, a_k, v_k)$$

**Image Editing Instruction** We employed image editing on training dataset to enhance the difficulty of inferring multi-modal samples. We maintain the original answer unchanged during the image editing process. We train a question generator $\mathcal{Q}$ to produce precise image edition instruction $I_v$ to control external image editing tools(*e.g.*, Flux.Kontext) for image editing. As we want to minimize the hallucination introduced in the process of question generation, we use judge model [51] ensures that the newly generated samples maintain consistency with the original ones, e.g. preserving the correctness of the answer after bootstrapping. we should maximize $\text{sim}(v_k, v_0)$ in semantics , where sim denotes consistency in the content of the problem investigation.

**Text Editing Instruction** Editing the original question and answer at the same time will also introduce a large number of hallucinations. We construct a text-editing instruction chain, where the instructions primarily involve performing text editing operations generated using external knowledge (e.g., mathematical concepts) on the answer from the previous step, thereby encouraging the inclusion of additional reasoning steps required to infer the final answer. We employ connective templates to seamlessly integrate the original multi-modal problem $(q, a, v)$ with the instruction $\mathcal{S}_k$.

For ground truth answer, we use the TTRL [16] to calculate the final answer. Specifically, for each function $f_i$ corresponding to the text editing instruction $I_i$ , we can get $a_k$:

$$a_k = \arg\max_{y \in \{y_1, \ldots, y_{N^k}\}} \sum_{i=1}^{N^k} \mathbb{I}\left[y_i = y\right], \quad y_i \sim \pi_\theta(\cdot \mid f_k\left(f_{k-1}(\ldots f_1(a_0)\ldots)\right)) \tag{4}$$

$\pi_\theta(\cdot \mid x)$ denotes the model's output distribution conditioned on the input $x$; $y_i \sim \pi_\theta(\cdot \mid x)$ indicates that the $i$-th output is sampled from the answer model's distribution conditioned on input $x$. To facilitate rapid convergence with minimal sampling while accounting for the unbounded nature of the solution space and the Markov property of action selection, we adopt the approximate formulation:

$$a_i = \arg\max_{y \in \{y_1, \ldots, y_N\}} \sum_{j=1}^{N} \mathbb{I}\left[y_j = y\right], \quad y_j \sim \pi_\theta(\cdot \mid f_i(a_{i-1})), \quad \forall i \in \{1, \ldots, k\} \tag{5}$$

Reducing time complexity from $\mathcal{O}(N^k)$ to $\mathcal{O}(kN)$, significantly accelerating large-scale sampling.

## 4.2 Progressive Instruction Evolution within LVLMs

In this section, we perform reinforcement learning training in both the question generator $\mathcal{Q}$ and the answer model $\mathcal{A}$. The final trained model $\mathcal{A}$ will serve as the model for reasoning. We alternate between training two models through Group Relative Policy Optimization [52] to achieve continuous growth in model capabilities.

**Answer Model Training** For the $i$-th iteration($i = 1, \dots$), we first train the answer model $\mathcal{A}$. We will train on $n$ data $X = (x_1, \dots, x_n) \in \mathcal{D}$ while using the question generator $\mathcal{Q}$ to generate the instruction $\mathcal{I}$. We use the answer model $A$ step by step to vote and generate the new answer $a_k$. Then we use the answer model $\mathcal{A}$ to generate a list of outputs $\{o_1, o_2, \dots, o_G\}$. We check whether answer is correct by using the rule-based mathruler [53] to directly judge whether the answer in response $o$ is same as $a_k$ functionally. As we want to maximize the probability the answer model get the true answer $\pi_\theta(a|q, v)$, we set reward as:

$$\mathcal{R} = \textbf{is\_equivalent}(a_k, o_i) + \textbf{format} \tag{6}$$

Where **format** refers to a reward score of format, which is designed to encourage the model to produce structured outputs, particularly those involving chain-of-thought reasoning.

**Question Generator Training** We then start to train the question generator $Q$. In each iteration, we sample training data $x \in \mathcal{D}$ from training dataset $\mathcal{D}$. As the sampled $k$ instruction $(I_1, \dots, I_k) \in \mathcal{I}$, we will use the question generator $\mathcal{Q}$ to generate a list of outputs $\{o_1, o_2, \dots, o_G\}$ which used to replace one instruction $I_{replaced} \in \{I_1, \dots, I_k\}$. We use the external image edition tool and answer model $A$ to get each augmented data $\{(q_1, a_1, v_1), \dots, (q_G, a_G, v_G)\}$. Due to limited answer model $\mathcal{A}$, we should also maximize the confidence level of the generated answers. The total confidence of the generated sample can be:

$$\mathcal{C} = \prod_{i=1}^{k} \text{sim}(v_i, v_{i-1}) \cdot \prod_{i=1}^{k} \pi_\theta(a_i | f_i(a_{i-1})) \cdot \textbf{format} \tag{7}$$

In the meantime, we want to maximize the difficulty for answer model $\mathcal{A}$, which minimizes the probability model $A$ can acquire the true answer $\pi_\theta(a_k | q_k, v_k)$ relative to original $\pi_\theta(a_0 | q_0, v_0)$. Therefore, the total reward can be defined as:

$$\mathcal{R} = \frac{\prod_{i=1}^{k} \text{sim}(v_i, v_{i-1}) \cdot \prod_{i=1}^{k} \pi_\theta(a_i | f_i(a_{i-1}))}{\pi_\theta(a_k | q_k, v_k) / \pi_\theta(a_0 | q_0, v_0)} \cdot \textbf{format} \tag{8}$$

In order to better calculate the Group Relative Advantages [52] and balance the relationship between scores of different magnitudes, we take the logarithm of all the scores and eliminate the common terms in the same group's outputs. To ensure a lower bound on performance, we take the maximum of the accuracy and the reciprocal of the number of evaluations, ultimately obtaining:

$$\mathcal{R}' = \sum_{i=1}^{k} \log \text{sim}(v_i, v_{i-1}) + \sum_{i=1}^{k} \log \pi_\theta(a_i | f_i(a_{i-1})) + \textbf{format}' - \log \max(\pi_\theta(a_k | q_k, v_k), \frac{1}{\textbf{test}}) \tag{9}$$

Where **test** denotes the number of evaluation trials used to assess accuracy, aligning with the number of responses generated in a reinforcement learning batch. The expression using max in the goal ensures at least one correct answer post-question augmentation, preventing variance collapse. The final question generator $\mathcal{Q}$ should be able to provide effective image editing without affecting the question, and be able to diversify the connection between the original problem and different instructions regarding the weaknesses of the answer model $\mathcal{A}$.

**Mixual Training** During training, it is essential to perform quality control on the instructions generated in each round. To ensure the reliability of the editing instructions, we employ a filtering strategy that rigorously eliminates instructions that do not conform to the required format or are prone to introducing hallucinations. In the meantime, to ensure stable training of model $\mathcal{A}$, the confidence $\mathcal{C}$

and the accuracy $\pi_\theta(a_k \mid q_k, v_k)$ of the generated questions should be maintained within a reasonable range. Reference to the design of Generative Adversarial Networks [54], to ensure the answer model $\mathcal{A}$ has enough capability, we train multiple steps with the answer model $\mathcal{A}$ and one step with the question generator $\mathcal{Q}$. This strategy is analogous to the way that training maintains answer model $\mathcal{A}$ train using data of appropriate difficulty, the $\{r_1, ..r_G\}$ for single sample's outputs $\{o_1, \ldots, o_G\}$ differ therefore model can maintain stable gradient updates during model training.

## 4.3 Enhanced Evolution with Progressive Training

To further promote the instructional evolution in the training process, it is essential to emphasize progressive training strategies to activate LVLMs' improved reasoning ability. Specifically, we gradually increase the number of sampled instructions $k$, guiding the model toward longer $k+1$-step reasoning chains in a curriculum learning manner. In each training round, the model learns to use $k$-step multi-modal chain-of-thoughts and receives process-level rewards on edited multi-modal data requiring larger $k$, thereby enhancing both the depth and accuracy of its reasoning. This step-wise progression enables the model to perform longer reasoning while maintaining correctness at each intermediate step, ultimately allowing it to solve complex multi-modal problems. In Algorithm 1, we present our training procedure pseudo code.

Throughout training, as the model's reasoning capabilities evolve, its confidence in executing editing instructions also increases. The framework progressively acquires the ability to support and perform a larger number of instruction edits with greater complexity. This continuous advancement forms a positive feedback loop: improved reasoning capabilities enable the execution of more sophisticated edits, which in turn expose the model to richer and more challenging training signals. Through this iterative refinement, the model gradually approaches its upper bound in reasoning performance. In Appendix D, we provide additional theoretical and experimental analyses on the stability and robustness of the evolved instruction generation process.

---

**Algorithm 1** EvolvedGRPO Training Procedure

---

**Require:** Initial policy $\pi_\theta$, training dataset $\mathcal{D}$, outer iterations Round, inner answer model training steps $T$, KL regularization weight $\beta$, curriculum step $c$, sample batch size $m$, GRPO iteration $\mu$
**Ensure:** Answer model policy $\pi_A^{Round}$
1: policy model $\pi_A^{(0)} \leftarrow \pi_\theta, \quad \pi_Q^{(0)} \leftarrow \pi_\theta$
2: **for** iteration $t = 0, \ldots, \text{Round} - 1$ **do**
3:     policy model $\pi_A^{(t+1)} \leftarrow \pi_A^{(t)}, \quad \pi_Q^{(t+1)} \leftarrow \pi_Q^{(t+1)}$
4:     editing instruction number $k \leftarrow \lfloor t/c \rfloor$
5:     **for all** sample $(q, v, a) \in \mathcal{D}$ **do**
6:         Generate edited sample $(q_k, v_k, a_k)$ through $\pi_A^{(t)}, \pi_Q^{(t)}$.
7:         Store $(q_k, v_k, a_k)$ into new dataset $\mathcal{D}^{(t)}$.
8:     **end for**
9:     **for** step $= 1, \ldots, T$ **do**
10:         Sample batch of $m$ samples $\{s^{(1)}, \ldots, s^{(m)}\} \sim \mathcal{D}^{(t)}$.
11:         Sample $G$ outputs $\{o_j\}_{j=1}^G$ for each sample $s_j$ and compute $A_j^l$ for the $l$-th token.
12:         **for** GRPO iteration $= 1, \ldots, \mu$ **do**
13:             Update the policy model $\pi_A^{(t+1)}$ with parameters $\theta_A^{(t+1)}$ using the GRPO gradient.
14:         **end for**
15:     **end for**
16:     Sample bath of $m$ samples $\{s^{(1)}, \ldots, s^{(m)}\} \sim \mathcal{D}^{(t)}$.
17:     Replace by $G$ editing instruction $\{o_j\}_{j=1}^G$ for the same instruction editing sample $s_j$.
18:     Compute $A_j^l$ for the $l$-th token through group relative advantage estimation.
19:     **for** GRPO iteration $= 1, \ldots, \mu$ **do**
20:         Update the policy model $\pi_Q^{(t+1)}$ with parameters $\theta_Q^{(t+1)}$using the GRPO gradient.
21:     **end for**
22: **end for**

---

# 5 EXPERIMENTS

## 5.1 Experimental Setup

**Datasets & Backbones.** As ideal data selections should be adaptable to diverse MLLM instruction datasets, we integrate DataTailor with one widely-used datasets to conduct experiments for its effectiveness evaluation: MMK12 [4] dataset, a high-quality and diverse multi-modal mathematical reasoning dataset, being composed of MAVIS [55], Geo3k [56], RCOT [57], MultiMath [58] datasets. For the general experimental setup, we adopt Qwen2.5-VL-7B-Instruct as the base model and train it using the GRPO strategy. More details of the experimental setup are shown in Appendix B.

**Benchmarks.** We evaluate model performance along two dimensions. First, we evaluate out-of-domain generalization across three visual reasoning benchmarks: MathVerse [59], MathVision [60], MathVista [61], and two textual reasoning benchmarks: GSM8K [62] and MATH500 [63]. Second, we evaluate the performance of EvolvedGRPO across general benchmarks, including MMMU [30], MMStar [31], and AI2D [64]. Additionally, we develop an evaluation suite to consistently assess our trained checkpoints and most open-source R1-related checkpoints using vLLM [65] for accelerated inference, while adopting reported results for other models. For parsing generated responses, we employ greedy decoding with GPT-4o [1] as the evaluation judge. Despite adhering closely to the system prompts from the original sources, slight deviations from reported results may persist, which we consider acceptable due to potential variations in judge models and evaluation procedures.

**Baselines.** We use the following baselines: (1) distilling knowledge from stronger external models (*e.g.*, GPT-4o, DeepSeek-R1), Mulberry [66] and Virgo [36] leverage distilled multi-modal chain-of-thought reasoning from more capable teacher models to conduct supervised fine-tuning, thereby enhancing the reasoning capabilities of smaller models; MindGym [10] improves reasoning capabilities by performing curriculum learning based chain-of-thought supervised fine-tuning on a set of multi-modal multi-hop questions synthesized by stronger teacher models; (2) collecting more challenging multi-modal perceptual datasets, OpenVLThinker [8] manually collects a large number of datasets with varying difficulty levels to support stage-wise reinforcement learning, thereby enabling curriculum-style learning guidance; NoisyRollout [14] increase data difficulty by enhancing visual perception; MM-EUREKA [4] facilitates model training by collecting large-scale data and dynamically filtering samples based on their difficulty at each training stage, thereby enabling the model to learn from appropriately challenging examples.

## 5.2 Main Results on Multi-modal Reasoning

We report the results of our EvolvedGRPO and other diverse methods shown in Table 1. Based on the observation of experimental results, we have summarized the following conclusions:

**Distillation-based methods exhibit a relatively low upper bound in enhancing reasoning capabilities.** We observe that in most benchmarks, Mulberry and Virgo exhibit limited reasoning capabilities by directly fine-tuning on distilled chain-of-thought data. Although MindGYM outperforms the open-sourced Qwen2.5VL-7B-Instruct by employing curriculum learning with synthesized multi-hop questions, it exhibits a limited upper bound of reasoning performance on long-chains benchmarks (*e.g.*, only 68.4 in MATH500) due to its reliance on externally generated questions. In contrast, our EvolvedGRPO achieves superior performance in all reasoning benchmarks owing to its progressively improved instructions.

**For collection-based approaches, the reasoning performances remain unsatisfactory due to the lack of diverse and complex data.** Although these baselines introduce improvements to GRPO-based training, they still underperform EvolvedGRPO in multimodal reasoning benchmarks due to their insufficient data-level enhancements to GRPO. OpenVLThinker and MM-EUREKA are constrained by the limited availability of human-annotated data, which prevents continuous data updates during the later stages of training, ultimately resulting in a significant performance gap. NoisyRollout augments the dataset with manually designed visual noise to increase perceptual difficulty, which forces the model to enhance its visual robustness. However, it still exhibits limited improvements in multimodal reasoning.

**Our EvolvedGRPO demonstrates strong reasoning capabilities.** Furthermore, in MathVision, the most difficult multimodal reasoning dataset, EvolvedGRPO attains the highest score (30.8), demonstrating its superior capability to handle complex multimodal reasoning tasks. Despite a slight

Table 1: Comparisons with our EvolvedGRPO and other baselines on multi-modal reasoning benchmarks. For fairness, all open-source reasoning frameworks are based on Qwen2.5-VL-7B-Instruct.

| Models | MathVista -mini | MathVision -full | MathVerse -mini | GSM8K | MATH500 | Avg. |
|---|---|---|---|---|---|---|
| GPT-4o | 63.8 | **36.8** | 50.2 | **94.2** | **74.6** | **63.9** |
| Mulberry-7B [66] | 63.1 | 22.8 | 39.6 | 69.1 | 65.2 | 52.0 |
| Virgo-7B [36] | 62.3 | 24.0 | 36.7 | 77.4 | 71.4 | 54.4 |
| MindGYM [10] | 70.3 | 28.6 | 48.4 | 84.1 | 68.4 | 60.0 |
| OpenVLThinker-7B [8] | 70.2 | 25.3 | 47.9 | 78.6 | 67.4 | 57.9 |
| NoisyRollout [14] | 72.9 | 28.9 | **52.8** | 79.0 | 68.4 | 60.4 |
| MM-EUREKA [4] | 73.0 | 26.9 | 50.3 | 82.5 | 66.8 | 59.9 |
| Qwen2.5-VL-7B-Instruct | 67.8 | 24.7 | 44.5 | 83.6 | 67.4 | 57.6 |
| **EvolvedGRPO (Ours)** | **74.0** | 30.8 | 51.8 | 85.1 | 73.2 | 63.0 |

performance gap with GPT-4o (63.0 vs. 63.9 on average), our method surpasses all other training methods based on Qwen2.5-VL-7B-Instruct, exhibiting a distinct advantage. It consistently improves upon the base model across all reasoning benchmarks, highlighting its robust reasoning ability.

## 5.3 Main Results on Downstream Generalization

Table 2 summarizes the results of our EvolvedGRPO and other baselines on downstream general benchmarks. Overall, our method generalizes well to unseen downstream tasks even without task-specific training, thanks to the improved multi-modal reasoning capabilities brought by our evolved instructions. However, on challenging benchmarks like MMMU, all existing reasoning approaches still fall short compared to GPT-4o, potentially due to the inherent gap between fine-grained perception

Table 2: Main results on general tasks.

| Models | MMMU | MMStar | AI2D |
|---|---|---|---|
| GPT-4o | 83.4 | 69.1 | 84.6 |
| Mulberry-7B [66] | 55.0 | 61.3 | 80.1 |
| Virgo-7B [36] | 46.7 | - | - |
| MindGYM [10] | - | 64.3 | - |
| OpenVLThinker-7B [8] | 51.9 | 63.2 | 82.7 |
| MM-EUREKA [4] | 52.3 | 64.1 | 81.4 |
| Qwen2.5-VL-7B-Instruct | 50.6 | 61.7 | 82.6 |
| **EvolvedGRPO (Ours)** | 53.0 | 62.7 | 83.3 |

and deep reasoning. We expect future work to explore richer instruction types that bridge this gap and enable reasoning grounded in fine-grained visual understanding.

## 5.4 In-depth Analysis

**Analysis of Model Training Paradigms** To investigate our EvolvedGRPO deeply, we study the ablation variants of different factors in Table 3. Specifically, we analyze their independence using the following ablation strategy: (1) Base model: we present the performance of the original Qwen2.5-VL-Instruct model as a baseline for comparison. (2) SFT : we perform Supervised Fine-Tuning on the base model using the K12 dataset for supervised learning. (3) GRPO: we apply Grouped Relative Policy Optimization on the base model using the K12 dataset for reinforcement learning. (4) ProgressiveGRPO: we build upon GRPO-based reinforcement learning by introducing curriculum-based dynamic instruction scaling on the K12 dataset, which continuously augments the training data with more instructions, without adversarial RL. (5) EvolvedGRPO: the method proposed in this work. It can be observed that: 1) **Direct SFT has limited effectiveness in enhancing the model's reasoning capabilities.** While direct fine-tuning via SFT improves performance on reasoning benchmarks (+4.9% overall), it fails to guide the model in developing better reasoning strategies. Furthermore, **RL methods also encounter bottlenecks during the scaling process.** GRPO demonstrates superior learning performance compared to SFT (+6.6% overall); however, it still encounters bottlenecks during the training process. 2) **The effectiveness of the GRPO algorithm can be enhanced through multi-stage guided data augmentation.** The progressive increase of edited instructions in a curriculum learning manner significantly enhances the effectiveness of reinforcement learning training, achieving improvements across all benchmarks (+8.9% overall). 3) **The introduction of adversarial reinforcement learning improves both instruction quality and the model's reasoning capabilities.** This iterative refinement leads to improved training efficiency and progressively stronger reasoning capabilities, ultimately eliciting the model's full reasoning potential(+12.1% overall).

Table 3: Results of ablation study to illustrate the effect of individual strategy.

| | Methods | MathVista ↑ | MathVision ↑ | MathVerse ↑ | GSM8k ↑ | MATH500 ↑ | Rel. ↑ |
|---|---|---|---|---|---|---|---|
| 1 | Base Model | 67.8 | 24.7 | 44.5 | 83.6 | 67.4 | 100.0% |
| 2 | SFT | 69.1 | 28.4 | 48.2 | 84.0 | 66.7 | 104.9% |
| 3 | GRPO | 71.9 | 28.9 | 49.6 | 82.9 | 67.0 | 106.6% |
| 4 | ProgressiveGRPO | 72.8 | 29.5 | 51.2 | 83.7 | 69.1 | 108.9% |
| 5 | EvolvedGRPO | **74.0** | **30.8** | **51.8** | **85.1** | **73.2** | **112.1%** |

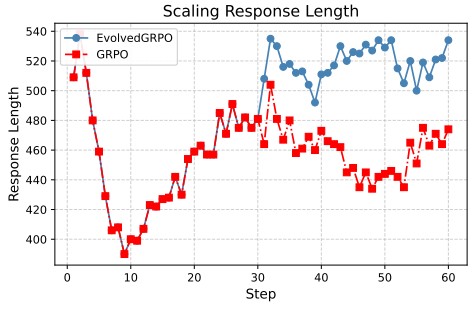

(a) Scaling Response Length

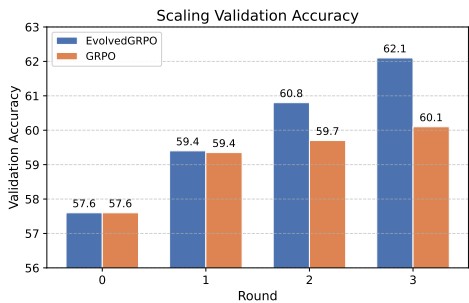

(b) Scaling Validation Accuracy

Figure 2: Comparison the Response Length and Validation Accuracy

**Scaling Visualization of Progressive Evolved Instructions.** In Figure 2, we illustrate the changes in reasoning chain length and model performance during training. Our method enables continuous growth in reasoning length. After each round of reinforcement learning, the model progressively adapts to longer reasoning chains, smoothly transitioning from $k$-step reasoning to $k+1$-step reasoning. In contrast, conventional RL-based methods tend to stabilize after the first round of adaptation, with reasoning lengths declining. As the reasoning length increases, our method consistently improves accuracy on the validation dataset. After each training round, the model successfully adapts to the current difficulty level while maintaining a reasonable accuracy rate. In comparison, the direct GRPO approach shows only marginal improvements beyond the second round, reflecting limited learning progress. We further provide a statistical analysis in Appendix C.

## 6  CONCLUSION

In this work, we propose EvolvedGRPO, a progressive instruction evolution framework that constructs increasingly challenging multi-modal reasoning datasets via evolved editing instructions to match the model's growing reasoning capabilities. We conceptualize the evolution of multi-modal reasoning instructions as multi-step editing across image and text modalities, and enhance it via adversarial rewards that jointly optimize the question generator and the answering model. Extensive experimental results demonstrate that our method progressively scales the instruction complexity without external supervision and continuously unlocks LVLMs' reasoning depth and adaptability across various tasks.

## Acknowledgements

This work was supported by the National Natural Science Foundation of China (62436007), the Key R&D Projects in Zhejiang Province (No. 2024C01106, 2025C01030), the NSFC (62272411), and the Zhejiang NSF (LRG25F020001), the Fundamental Research Funds for the Central Universities (226-2025-00017), Ningbo Yongjiang Talent Introduction Programme (2024A-401-G), Zhejiang University Education Foundation Qizhen Scholar Foundation.

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

# Appendix

Our appendix is organized as follows.

- More Theoretical Analysis on EVOLVEDGRPO (Section A).
- Experimental details of our EVOLVEDGRPO (Section B).
- External Effectiveness Analysis of EVOLVEDGRPO (Section C).
- Detailed Evolution Illustration of EVOLVEDGRPO Training Pipeline (Section D).
- Qualitative examples of EVOLVEDGRPO and BASE MODEL (Section E).
- Limitations of EVOLVEDGRPO (Section F).

## A  More Theoretical Analysis on EvolvedGRPO

### A.1  Problem Setup

We provide a theoretical analysis of the convergence behavior of EvolvedGRPO, which alternates training between the question generator policy $\pi_Q$ and the answer model policy $\pi_A$ under the Grouped Relative Policy Optimization (GRPO) framework.

We next prove that the adversarial training process is convergent for any finite number of $k$ instruction editing steps, which allows us to generalize the convergence to the entire model training process.

Let $\pi_Q$ be the policy that generates a sequence of instructions $I = (I_1, \ldots, I_k)$ for given sample $s = (q_0, a_0, v_0)$. Let $\pi_A$ be the answer model policy generating answer $a$ conditioned on multimodal input $(q, v)$. The expected total reward objective is:

$$\mathcal{J}(\pi_A, \pi_Q) = \mathbb{E}_{(q_k, v_k, a_k)} \left[ \mathcal{R}'(\pi_A; q_k, v_k, a_k) \right] \tag{10}$$

where the reward function $\mathcal{R}'$ is defined as:

$$\mathcal{R}' = \textbf{format}' + \sum_{i=1}^{k} \log \text{sim}(v_i, v_{i-1}) + \sum_{i=1}^{k} \log \pi_A(a_i | f_i(a_{i-1})) - \log \max \left( \pi_A(a_k | q_k, v_k), \frac{1}{\textbf{test}} \right) \tag{11}$$

The objective of EvolvedGRPO is to solve the following min-max problem:

$$\min_{\pi_A} \max_{\pi_Q} \mathcal{J}(\pi_A, \pi_Q). \tag{12}$$

When optimizing the answer model policy $\pi_A$, the training sample $(q_k, v_k, a_k)$ is fixed, making most reward components fixed to $\pi_A$. Specifically, the term: $\mathbb{E}_{(q_k, v_k, a_k)} \left[ \sum_{i=1}^{k} \log \text{sim}(v_i, v_{i-1}) + \sum_{i=1}^{k} \log \pi_A(a_i | f_i(a_{i-1})) \right]$ is decided by old policy $\pi_{A_{old}}$ and $\pi_{Q_{old}}$, being independent of current policy $\pi_A$, and can be treated as a constant during training. Therefore, the reward function effectively reduces to maximize $\mathbb{E}_{(q_k, v_k, a_k)} \left[ \pi_A(a_k | q_k, v_k) \right]$. This simplification allows us to focus the GRPO update on maximizing the likelihood of the final answer token under the current instruction-conditioned policy. All preceding terms serve only as context and baseline for shaping the total reward but do not influence gradients with respect to $\pi_A$.

### A.2  Proof of Convergence for GRPO

**Theorem A.1** (Convergence of GRPO). *Let $\{\pi_{\theta_t}\}_{t=0}^{\infty}$ be the sequence of policies updated using GRPO. Then, Then, under the standard assumption that each update approximately maximizes the GRPO objective, the policy sequence $\pi_{\theta_t}$ converges to a stationary policy $\pi^*$.*

*Proof.* Let $\eta(\pi) = \mathbb{E}_{x \sim \mathcal{D}, y \sim \pi(\cdot|x)}[R(x, y)]$ denote expected return of policy $\pi$ and we have:

$$\mathcal{J}_{\text{GRPO}}(\theta; \theta_t) = \mathbb{E}_{x \sim \mathcal{D}, y \sim \pi_{\theta_t}(\cdot|x)} \left[ \min \left( r A(x, y), \text{clip}(r, \epsilon) A(x, y) \right) - \beta \cdot D_{\text{KL}}(\pi_\theta \| \pi_{\theta^t}) \right], \tag{13}$$

where $r$ is importance sampling ratio $\frac{\pi_\theta(x, y)}{\pi_{\theta_t}(x, y)}$, and $A(x, y)$ is the normalized group-based advantage.

Since each update approximately maximizes $J_{\text{GRPO}}(\theta)$, we have:

$$\mathcal{J}_{\text{GRPO}}(\theta_{t+1}; \theta_t) \geq \mathcal{J}_{\text{GRPO}}(\theta_t; \theta_t). \tag{14}$$

Following the standard policy improvement argument, the change in return satisfies:

$$\eta(\pi_{\theta_{t+1}}) - \eta(\pi_{\theta_t}) = \mathbb{E}_{x \sim \mathcal{D}} \left[ \mathbb{E}_{y \sim \pi_{\theta_{t+1}}(\cdot|x)}[\mathcal{R}(x, y)] - \mathbb{E}_{y \sim \pi_{\theta_t}(\cdot|x)}[\mathcal{R}(x, y)] \right] \tag{15}$$

$$= \mathbb{E}_{x \sim \mathcal{D}} \left[ \mathbb{E}_{y \sim \pi_{\theta_t}(\cdot|x)} \left[ \frac{\pi_{\theta_{t+1}}(y|x)}{\pi_{\theta_t}(y|x)} \cdot \mathcal{R}(x, y) \right] - \mathbb{E}_{y \sim \pi_{\theta_t}(\cdot|x)}[\mathcal{R}(x, y)] \right] \tag{16}$$

$$= \mathbb{E}_{x \sim \mathcal{D},\, y \sim \pi_{\theta_t}(\cdot|x)} \left[ \left( \frac{\pi_{\theta_{t+1}}(y|x)}{\pi_{\theta_t}(y|x)} - 1 \right) \cdot \mathcal{R}(x, y) \right] \tag{17}$$

$$\geq \mathbb{E}_{x \sim \mathcal{D},\, y \sim \pi_{\theta_t}(\cdot|x)} \left[ \log \frac{\pi_{\theta_{t+1}}(y|x)}{\pi_{\theta_t}(y|x)} \cdot \mathcal{R}(x, y) \right]. \tag{18}$$

Since GRPO includes an explicit KL penalty term via clipping, we can apply the **Pinsker inequality**:

$$D_{\text{KL}}(\pi_{\theta_t} \| \pi_{\theta_{t+1}}) \geq \frac{1}{2} \| \pi_{\theta_{t+1}} - \pi_{\theta_t} \|_1^2. \tag{19}$$

Combine the definition of equation 13, inequality 14 and inequality 19, we can get:

$$\eta(\pi_{\theta_{t+1}}) - \eta(\pi_{\theta_t}) \geq \frac{\beta}{2} \| \pi_{\theta_{t+1}} - \pi_{\theta_t} \|_1^2. \tag{20}$$

As $\eta(\pi_{\theta_t})$ is upper bounded (since $R(x, y) \in [R_{min}, R_{max}]$) and non-decreasing according to inequality 15, it converges to a limit $\eta^*$. Thus, we have:

$$\lim_{t \to \infty} \| \pi_{\theta_{t+1}} - \pi_{\theta_t} \|_1 = 0. \tag{21}$$

Since the space of probability distributions over discrete outputs is compact under the $L_1$ norm, it follows that the sequence $\pi_{\theta_k}$ converges to a stationary policy $\pi^*$.

$\square$

### A.3 Proof of Convergence for EvolvedGRPO

**Definition A.1** (Local Nash Equilibrium). *A pair of policies $(\pi_Q^*, \pi_A^*)$ is a local Nash equilibrium of $J(\pi_Q, \pi_A)$ if there exist neighborhoods $\mathcal{N}_Q$ and $\mathcal{N}_A$ such that:*

$$\mathcal{J}(\pi_A^*, \pi_Q^*) \leq \mathcal{J}(\pi_A, \pi_Q^*) \quad \forall \pi_A \in \mathcal{N}_A, \tag{22}$$
$$\mathcal{J}(\pi_A^*, \pi_Q^*) \geq \mathcal{J}(\pi_A^*, \pi_Q) \quad \forall \pi_Q \in \mathcal{N}_Q. \tag{23}$$

Since the two policies sequentially optimize their respective objectives, we have:

$$\mathcal{J}(\pi_A^{(t+1)}, \pi_Q^{(t)}) \leq \mathcal{J}(\pi_A^{(t)}, \pi_Q^{(t)}) \tag{24}$$
$$\mathcal{J}(\pi_A^{(t+1)}, \pi_Q^{(t+1)}) \geq \mathcal{J}(\pi_A^{(t+1)}, \pi_Q^{(t)}) \tag{25}$$

As we use $D_{\text{KL}}[\pi_\theta \| \pi_{\text{ref}}] = \frac{\pi_{ref}(o_{i,t}|q, o_{i<t})}{\pi_\theta(o_{i,t}|q, o_{i<t})} - \log \frac{\pi_{ref}(o_{i,t}|q, o_{i<t})}{\pi_\theta(o_{i,t}|q, o_{i<t})} - 1$ to prevent the model from forgetting previously acquired knowledge, while the answer model has the capability to master the reasoing skill in $k$-step reasoning through GRPO training. We can make the following assumption:

**Assumption 1** (Answer Model Consistence). *The answer model exhibits monotonic improvement in reasoning performance for all editing instructions generated by the question generator during training. Formally, for any iteration $t$,*

$$\forall \pi_Q, \quad \mathcal{J}(\pi_A^{(t+1)}, \pi_Q) \leq \mathcal{J}(\pi_A^{(t)}, \pi_Q). \tag{26}$$

Table 4: Training hyper-parameters of EvolvedGRPO.

| Hyper-parameters | Question Generator | Answer Model |
|---|---|---|
| LLM Init | Qwen2.5-7B-VL-Instruct | Qwen2.5-7B-VL-Instruct |
| KL Penalty | Low Variance KL | Low Variance KL |
| KL Coefficient | $1 \times 10^{-2}$ | $1 \times 10^{-2}$ |
| Optimizer | AdamW | AdamW |
| Learning Rate | $1 \times 10^{-6}$ | $1 \times 10^{-6}$ |
| Weight Decay | $1 \times 10^{-2}$ | $1 \times 10^{-2}$ |
| Numerical Precision | AMP32 | AMP32 |
| Gradient Clipping | 1.0 | 1.0 |
| Rollout $n$ | 5 | 5 |
| Rollout Temperature | 1.0 | 1.0 |
| Rollout Top-$p$ | 0.99 | 0.99 |
| Rollout Batch Size | 512 | 512 |
| Micro Batch Size for Update | 16 | 16 |
| Micro Batch Size for Experience | 64 | 64 |
| Training Steps | 1 | 30 |
| Resource Usage | $4\times$ RTX A6000 | $4\times$ RTX A6000 |
| Total Epochs | | 5 |

Therefore, the accuracy of the answer model is guaranteed to be non-decreasing as different instruction editions are applied. Consider the space of $k$-step instruction editing strategies is finite, while Theorem A.1 suggested the convergence of the answer model under the condition of finite $k$-step edition training datasets, the final answer model can also converge. In this situation, the final answer model cannot improve its performance under $\pi_Q^*$ by changing its policy $\pi_A^*$. Furtheremore, we have:

$$\forall \pi_Q, \quad \lim_{t \to \infty} \pi_A \to \pi_A^*(\pi_Q) \tag{27}$$

During the evolution of policy $\pi_Q$ and $\pi_A$ consistently converges to its optimal mode. Then we only consider the policy $\pi_Q$ under the final condition $\pi_A^*$, according to the Theorem A.1, the $\pi_Q$ also converges to $\pi_Q^*$ therefore maximize $\mathcal{J}(\pi_A^*, \pi_Q)$. Finally, $(\pi_A, \pi_Q)$ converge to $(\pi_A^*, \pi_Q^*)$ which is Local Nash Equilibrium. That is, the final equilibrium corresponds to a state where the answer model is robust, and the questions are maximally adversarial, yet no longer effective in reducing accuracy.

## B Experimental Details

### B.1 Detailed Training Hyperparameters

We train our models using GRPO strategy. The both models are initialized with Qwen2.5-VL-7B-Instruct [23]. The detailed hyper-parameters used during training are summarized in Table 4. In the training process, the details of two different types of instructions are presented in Table 5.

As shown in Table 6, we report the time cost of each module in our training pipeline. To reduce the overhead introduced by data augmentation, we adopt the vLLM framework for efficient and parallelized inference, which significantly accelerates the instruction editing process. Compared to directly training with the raw model dataset, our method incurs only an additional 10.0% computational cost.

To enhance the model's reasoning capability, we use the template "You FIRST think about the reasoning process as an internal monologue and then provide the final answer. The reasoning process MUST BE enclosed within <think> </think> tags. The final answer MUST BE put in \boxed{}." for answer model during both training and evaluation. During reward computation, a format reward is calculated, and only the content enclosed within "\boxed{}" is considered as the definitive answer.

Table 5: Comparison of representative methods for image and text editing.

| Attribute | Image Editing | Text Editing |
|-----------|---------------|--------------|
| Input Type | Image + Instruction | Text + Instruction |
| Output Type | Image | Text |
| Editing Target | Pixel-level changes | Sentence Addition |

Table 6: Resource cost statistics for each stage of the training pipeline using $4\times$ RTX A6000. Stage 1 involves question generation, Stage 2 trains answer model, and Stage 3 trains question generator.

| Training Stage | Training Time (hours) |
|----------------|-----------------------|
| Stage 1: Instruction Generation | 4 |
| Stage 2: Answer Model Training | 60 |
| Stage 3: Question Generator Training | 2 |

## B.2 Training Dataset Description

**MMK12** encompasses mathematical problems across various knowledge domains, including geometry, functions, spatial reasoning, and more. Some important categories are introduced as follows:

Function Reasoning: This task requires models to understand function concepts, analyze function graphs and expressions, and apply function properties to solve problems. This type of reasoning develops the model's ability to comprehend abstract mathematical concepts, fostering its capability to identify function characteristics, determine critical points, and analyze function behavior.

Geometric Reasoning: This task involves applying spatial relationships, geometric theorems, and properties of shapes. Through geometric reasoning training, models enhance their spatial visualization, logical deduction, and formalization abilities for geometry problems, enabling them to solve complex problems in both plane and solid geometry.

Pattern Reasoning: This type of task focuses on understanding flow diagrams and recognizing patterns in visual sequences. Models need to discover patterns, predict rule-based changes, or understand logical relationships in visual content. This task examines the model's pattern recognition abilities, inductive reasoning skills, and visual logical thinking.

Table 7: Comparison of datasets in terms of scope, type, image source, QA source, CoT answer source, and dataset size.

| Dataset | Scope | Type | Img. Source | QA Source | CoT Answer Source | Size |
|---------|-------|------|-------------|-----------|-------------------|------|
| MAVIS | Geo & Func | MCQ & FB | Synthetic | Synthetic Engine | GPT-4o | 20K |
| Geo3k | Geo | FB | Real World | Real World | None | 3K |
| RCOT | Geo | MCQ & FB | Synthetic | Synthetic Engine | GPT-4o | 10K |
| MultiMath | Diverse | MCQ & FB | Real World | GPT-4o | GPT-4o | 6.4K |
| MMK12 [4] | Diverse | FB | Real World | Real World | Real World | 15.6K |

## B.3 Evaluation Details

To further assess the reasoning capabilities of the models, we construct a validation set comprising 3,000 instances, sampled from multiple benchmarks with 500 instances drawn from each. Detailed information about the benchmarks is provided in Table 8. The results of the validation experiments are reported in Section 5.4 and Appendix D.

## C    Effectiveness Analysis of EvolvedGRPO

In this section, we discuss the effectiveness of EvolvedGRPO for improving the reasoning of LVLMs with low overhead.

Table 8: Summary of benchmarks for multimodal and mathematical reasoning evaluation.

| Benchmark | Task Description | Modality | Size | Source |
|---|---|---|---|---|
| MathVista-mini | Visual math reasoning | Image+Text | 1.0K | MathVista |
| MathVision-full | Diagram-based math QA | Image+Text | 3.0K | MATH-Vision |
| MathVerse-mini | Multiformat math QA | Image+Text | 3.1K | MathVerse |
| GSM8K | Grade-school math QA | Text | 8.5K | OpenAI |
| MATH500 | Competition-level math QA | Text | 500 | MATH Dataset |
| ValidationTest | Math QA | Multi-modal | 3.0K | - |

Table 9: Effectiveness of each equation.

| | Methods | MathVista ↑ | MathVision ↑ | MathVerse ↑ | GSM8K ↑ | MATH500 ↑ | Rel. ↑ |
|---|---|---|---|---|---|---|---|
| 1 | **EvolvedGRPO (Ours)** | **74.0** | 30.8 | **51.8** | **85.1** | **73.2** | **112.1%** |
| 2 | w/o Image Edition | 73.4 | 29.8 | 51.3 | 84.5 | 72.5 | 110.6% |
| 3 | w/o Text Edition | 72.1 | **31.8** | 50.9 | 81.3 | 68.7 | 109.7% |
| 4 | w/o Image Edition & Text Edition | 71.9 | 28.9 | 49.6 | 82.9 | 67.0 | 106.6% |
| 5 | Qwen2.5-VL-7B-Instruct | 71.9 | 28.9 | 49.6 | 82.9 | 67.0 | 106.6% |

Table 10: Avg performance of all domains on EvolvedGRPO disciplinary reasoning.

| | Models | Physics ↑ | Chemistry ↑ | Biology ↑ | Avg. ↑ |
|---|---|---|---|---|---|
| 1 | **EvolvedGRPO (Ours)** | **61.5** | **65.5** | 63.9 | **63.6** |
| 2 | OpenVLThinker | 53.8 | 60.6 | 65.0 | 59.8 |
| 3 | MM-Eureka | 56.2 | 65.2 | **65.2** | 62.2 |
| 4 | Qwen2.5-VL-7B-Instruct | 45.4 | 56.4 | 54.0 | 51.9 |
| 5 | + GRPO | 51.6 | 57.1 | 58.6 | 55.8 |
| 6 | + EvolvedGRPO w/iter 1 | 51.4 | 56.4 | 57.2 | 55.0 |
| 7 | + EvolvedGRPO w/iter 2 | 54.2 | 60.8 | 57.6 | 57.5 |

## C.1 Effectiveness Analysis

To examine the effectiveness of each type of instruction, we first remove the image editing instructions and text editing instructions separately, and observe the change in model performance after training. The complete results are presented in Table 9. We can find that both text editing instruction (+4.0%) and image editing instruction (+3.1%) are effective for EvolvedGRPO.

## C.2 Experimental Results in a Wider Range of Fields

To demonstrate the framework's generality and effectiveness, we leverage Qwen2.5-VL-7B-Instruct to perform complex editing instructions across various vision-language tasks. EvolvedGRPO continues to demonstrate its effectiveness across multiple real-world domains such as physics, chemistry, and biology. The results in Table 10 provide strong evidence that our method effectively decouples visual and textual information in diverse areas, leading to a significant enhancement of the model's comprehensive reasoning abilities.

Furthermore, the gradual improvement in model performance during the first two rounds of training in Table 10 also demonstrates the effectiveness of multi round data augmentation training.

## D   Detailed Evolution Illustration of EvolvedGRPO Training Pipeline

## D.1   Theoretical Analysis

For training data that the model has repeatedly encountered, the accuracy can approach $100\%$ over multiple learning iterations for part of the data. In such cases, since each evaluation samples only limited responses and the reward is variance-normalized, repeated training causes the responses to converge. When all responses are correct, the variance becomes 0 and is thus ignored, leading to

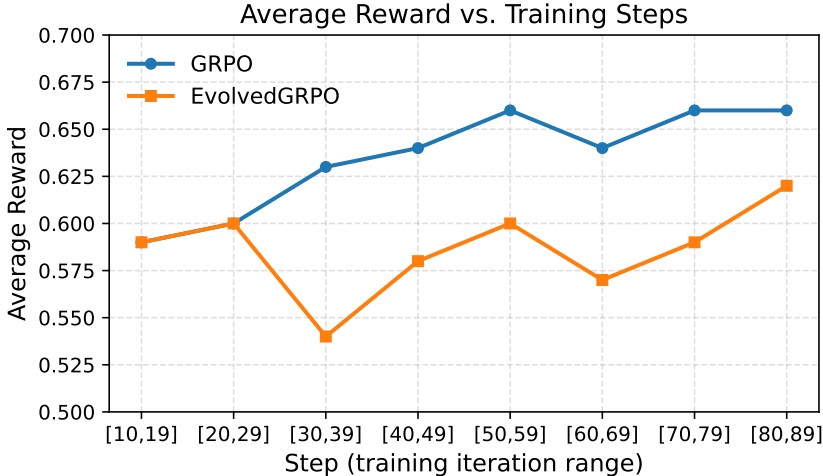

Figure 3: Average reward for the corresponding intervals.

training stagnation. Furthermore, in order to increase accuracy, the model may omit the reasoning process entirely or produce an incorrect reasoning process that nevertheless yields the correct answer, resulting in negative rewards that can cause training collapse.

In contrast, EvolvedGRPO introduce additional informational complexity during multi-round training, thereby increasing the data's informational entropy and increasing reasoning complexity. This process reduces the model's accuracy, strategically promoting an increase in cognitive load that drives the model to engage in more nuanced multi-step inferential reasoning, preventing training stagnation and fostering further reasoning learning. Furthermore, the integration of additional information and the escalation of difficulty within the synthetic data forces the model to extend its reasoning processes, impeding direct answer memorization and reliance on spurious memorization of erroneous inferential paths and thereby mitigating the risk of training collapse.

## D.2  Experimental Analysis

In this part we calculate the average reward for the corresponding intervals in Figure 3.

We observe that EvolvedGRPO exhibits greater fluctuations during each training round, as it can consistently improve after data augmentation, thereby maintaining a stable average performance without rapid decline. Specifically, we found that the reward in EvolvedGRPO fluctuates continuously, with the average reward sharply decreasing after each data augmentation before gradually increasing. In contrast, the reward growth in GRPO slows down and eventually stagnates.

The analysis confirms that by generating more complex problems, EvolvedGRPO allows for a richer variety of high-quality solutions, indicating a more diverse and nuanced reward landscape that encourages exploration over exploitation.

## D.3  Training Visualization

Since reward signals in multi-modal reasoning are closely tied to the depth and structure of the reasoning process, changes in entropy at both the visual and textual semantic levels can effectively modulate the reward landscape. Specifically, altering the number of inference steps—particularly in extended reasoning chains involving over $4096$ tokens—tends to induce more substantial and informative reward variations than shallow input perturbations.

In this section, we report the detailed evolution process of the first three rounds of EvolvedGRPO in Figure 4. Additionally, the number of editing steps is increased once the reward stabilizes and the length of the reasoning chain remains unchanged, allowing for further difficulty scaling.

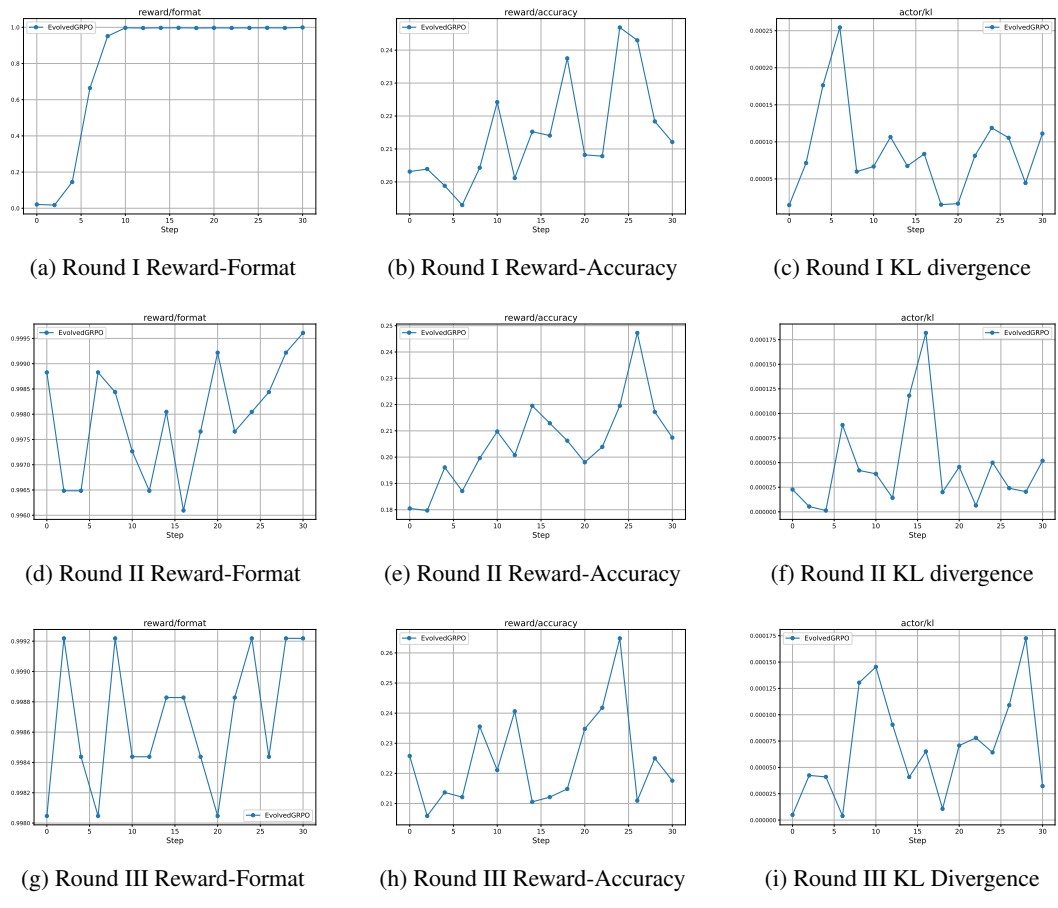

Figure 4: Detailed Evolution Illustration of Reward-Accuracy, Reward-Format, KL Divergence for the answer model over Round I, II, and III.

# E Qualitative Examples

In Figure 5, we demonstrate the advantages of EvolvedGRPO in complex reasoning problems such as geometry and statistics.

# F Limitations

Due to the limited capabilities of current image editing tools, our approach to image manipulation remains relatively simple. In future work, we plan to explore hybrid training of LVLMs with more advanced image editing tools. This integration is expected to further enhance the model's reasoning ability and enable more complex and realistic image transformations.

**Question:**
Please answer the question and provide the correct option letter, e.g., A, B, C, D, at the end.
As shown in the figure, P is a point outside ⊙O, PA and PB intersect ⊙O at two points C and D respectively. It is known that the central angles of ⌢AB and ⌢CD are 90.0 and 50.0 respectively, then ∠P = (   )
(A) 45°
(B) 40°
(C) 25°
(D) 20°

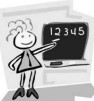

**Answer:** *D*

---

**Qwen2.5-VL-7B:**
*To solve this problem, we need to use the properties of angles in a circle and the relationship between the central angles and the inscribed angles.*
*1. **Identify the given information:**\n   - The central angle of arc \\( \\overset{\\frown}{AB} \\) is \\( 90^\\circ \\).\n   - The central angle of arc \\( \\overset{\\frown}{CD} \\) is \\( 50^\\circ \\).*
*2. **Use the exterior angle theorem:**\n   - The exterior angle \\( \\angle P \\) is the sum of the two inscribed angles \\( \\angle ADB \\) and \\( \\angle ACD \\):\n   \\[\n   \\angle P = \\angle ADB + \\angle ACD = 45^\\circ + 25^\\circ = 70^\\circ\n . .*
*3. **Correct approach:**   - The angle \\( \\angle P \\) is the difference between the angles formed by the tangents and the chords:\n   \\[\n   \\angle P = 90^\\circ - 50^\\circ = 40^\\circ\n*
*Option: Therefore, the correct answer is: B*

---

**EvolvedGRPO:**
<think>
To solve for the angle ∠P, we need to use the properties of the angles in a circle and the relationships between the central angles and the inscribed angles.\n\n1. The central angle of arc AB is 90°, so the inscribed angle ∠ACB (which is half of the central angle) is 45°.\n2. The central angle of arc CD is 50°, so the inscribed angle ∠CAD (which is half of the central angle) is 25°.\n3. Since ∠ACB and ∠CAD are angles in the same segment, and ∠P is the external angle to these two angles, we can use the fact that the external angle of a triangle is equal to the sum of the two non-adjacent internal angles.
Thus, ∠P = ∠ACB - ∠CAD = 45° - 25° = 20°.
</think>
*Option: D*

---

**Question:**
Mary has written all the numbers from 1 to 30 . How many times has she written digit 2?

**Answer:** 13

---

**Qwen2.5-VL-7B:**
*To determine how many times the digit 2 appears in the numbers from 1 to 30, we need to count the occurrences of the digit 2 in each place value (units and tens).\n\n1. **Count the digit 2 in the units place:**\n   - The numbers that have 2 in the units place are: 2, 12, 22, 32.\n   - There are 4 such numbers.\n\n2. **Count the digit 2 in the tens place:**\n   - The numbers that have 2 in the tens place are: 20, 21, 22, 23, 24, 25, 26, 27, 28, 29.\n   - There are 10 such numbers.\n\nAdding these together, the total number of times the digit 2 appears is:\n\\[ 4 + 10 = 14 \\]\n\nSo, Mary has written the digit 2 a total of 14 times.*

*Option: 14*

---

**EvolvedGRPO:**
<think>
To determine how many times the digit 2 appears in the numbers from 1 to 30, we need to consider the occurrences of the digit 2 in both the tens place and the units place.\n\n1. **Units place**: The digit 2 will appear in the units place in the numbers 2, 12, 22. So, it appears 3 times.\n2. **Tens place**: The digit 2 will appear in the tens place in the numbers 20, 21, 22, 23, 24, 25, 26, 27, 28, 29. So, it appears 10 times.\n\nAdding these together, the digit 2 appears \\(3 + 10 = 13\\) times.
</think>

*Option: 13*

---

Figure 5: Example of complex reasoning. Red highlights Qwen2.5-VL-7B-Instruct's reasoning errors, Green shows EvolvedGRPO-7B's correct answer.

