# OpenReview forum: "EvolvedGRPO: Unlocking Reasoning in LVLMs via Progressive Instruction Evolution"
_NeurIPS.cc/2025/Conference — NeurIPS 2025 poster_

### Official Review · Reviewer_moty · 2025-06-24

**Clarity:** 1
**Significance:** 2
**Originality:** 3
**Rating:** 3
**Confidence:** 4

**Summary:**

This paper aims to address the problem of training stagnation caused by reward convergence when fine-tuning LVLMs with RL methods like GRPO. To tackle this challenge, the authors propose EvolvedGRPO, a progressive instruction evolution framework. This framework utilizes an adversarial setup where a question generation model (Q) dynamically creates training samples of increasing difficulty for an answer model (A). Specifically, model Q generates image and text editing instructions to increase the complexity of problems, thereby matching the evolving capabilities of model A. Ultimately, experimental results show that this method achieves state-of-the-art performance on several multi-modal reasoning benchmarks.

**Questions:**

See Weaknesses

**Ethical Concerns:**

["NO or VERY MINOR ethics concerns only"]

**Final Justification:**

I still think that the paper lacks clarity in its methodological description.

Additionally, the central claim that the method solves reward convergence and training stagnation remains unsubstantiated. The evidence provided in the rebuttal is unconvincing, and a promise of future analysis is insufficient at this stage.

Due to these fundamental issues, I will maintain my original rating.

**Limitations:**

yes.

**Quality:**

2

**Strengths And Weaknesses:**

Strengths:
1. Impressive Performance: The paper achieves excellent results on several challenging multi-modal reasoning benchmarks, outperforming other open-source models and even approaching the performance of the closed-source GPT-4O on certain metrics.
2. Compelling Motivation: The paper's premise is highly valuable. Exploring self-evolving reinforcement learning algorithms to dynamically adjust training data difficulty is an important and promising research direction for pushing model capabilities beyond their current limits and achieving a higher level of intelligence.

Weaknesses:

1. Lack of Clarity and Missing Key Details: For a paper whose core contribution is a new algorithm, the description of the method is surprisingly scattered and vague. It dedicates considerable space to well-known background concepts like SFT and GRPO, which hinders the understanding and reproducibility of its core contribution.

    - In the core reward function, Equation (9), the symbol format' appears without any definition or explanation provided anywhere in the paper. Additionally, the explanation for the test variable is also same problem.

    - The "Text Editing Instruction" is a central innovation of the method, yet the main text lacks a detailed description of its specific operations. Readers are forced to infer its meaning from the example in Figure 1, but such a key mechanism should be explicitly and thoroughly described in the text.
    - The paper's description of the adversarial training process is not systematic. A clear algorithm flowchart or pseudocode block is crucial for understanding the alternating training of the Q-A models and key steps like instruction generation and replacement, but this is absent from the main body of the paper.
    - The presentation of results in Table 1 could be misleading. Specifically, the 'Avg.' column is calculated and presented even for baseline models where scores for the GSM8K and MATH500 datasets are absent (e.g., OpenVLThinker-7B, NoisyRollout, MM-EUREKA). This creates an inconsistent basis for comparison and could cause readers to misinterpret the overall performance of these baseline models


    - Typo: A typo, "generation generation," appears on line 195, which affects the reading experience.


2. Mismatch Between Motivation and Method's Effect: The paper's initial motivation is to solve the RL training bottleneck in multi-modal reasoning caused by the entanglement of visual and textual information. However, its core "text editing" method—for instance, transforming an answer z into a problem of solving for D = 3*z/4 + 9/4—essentially appends a separate, unimodal mathematical calculation task at the end of the original multi-modal task. This operation does not increase the intrinsic complexity of the visual-textual reasoning itself, but rather appears to train the model's arithmetic capabilities.

3. The paper lacks the necessary analysis to prove that this method actually solves the problem it set out to address: reward convergence and training stagnation. For example, the authors do not analyze whether the reward variance for different responses to the same sample actually increases after using EvolvedGRPO. The logical link between the method's effect and the initial motivation is not established.
Insufficient Experimental Analysis and Comparison:

4. The paper's image manipulation method (e.g., adding noise or filters) is conceptually very similar to that of NoisyRollout[1]. However, the paper merely cites this work without providing any in-depth comparative analysis or discussing the differences in concept or effectiveness.
The paper lacks a dedicated ablation study for the image editing component. From the experiments, we cannot determine the actual performance contribution of adding noise or editing the images. This calls into question the effectiveness of the "multi-modal" aspect of the instruction evolution.

[1] NoisyRollout: Reinforcing Visual Reasoning with Data Augmentation

---

> ### Author Rebuttal · Authors · 2025-07-31
>
> We sincerely thank the reviewers for their exceptionally thorough and detailed comments!  We will explain your concerns point by point.
>
> > Q1: In the core reward function, Equation (9), the symbol format' appears without any definition or explanation provided anywhere in the paper. Additionally, the explanation for the test variable is also same problem.
>
> A1: Thank you for pointing this out.
>
> `format` refers to a reward score of format, which is designed to encourage the model to produce structured outputs, particularly those involving chain-of-thought reasoning.
>
> `test` denotes the number of evaluation trials used to assess accuracy, aligning with the number of responses generated in a reinforcement learning batch. The expression using `max` in the goal ensures at least one correct answer post-question augmentation, preventing variance collapse.
>
> We will include a more detailed description in the revised version to improve accessibility and clarity.
>
> > Q2: Clarification of Multi-modal Instruction Editing.
>
> A2: We clarify that the core contribution of EvolvedGRPO lies in its formulation of a model-agnostic, multimodal, self-evolving reinforcement learning framework applicable across diverse domains (e.g., physics, chemistry) using various tools while continuously improving through self-enhancement. Therefore, we emphasize the overall self-evolving pipeline rather than any domain-specific instantiation. For mathematical tasks, EvolvedGRPO adaptively selects appropriate modules for generating text editing instructions and employs external tools (e.g., Prisma Art, OpenCV, and Neural Style) to increase the visual complexity of mathematical images. This part was included in the appendix for brevity; however, we will incorporate a detailed description of the specific tools into the main text in the revised version.
>
> > Q3: A clear algorithmic flowchart or pseudocode is currently missing from the main body of the paper.
>
> A3: The algorithm flowchart is currently included in Appendix A.4: PseudoCode (located in Supplementary Material.zip) due to space constraints. It will be moved to the main body of the paper in the revised version.
>
> > Q4: The presentation of results in Table 1 could be misleading.
>
> A4: In earlier evaluations, we cited results from original papers due to the lack of standardized evaluation protocols. To ensure fairness and provide a comprehensive comparison, we have carefully re-evaluated the results using consistent prompts across all models. The updated results are displayed below:
>
> Table 1: Expanded version of the main table
>
> | Models                 | MathVista-mini | MathVision-full | MathVerse-mini | GSM8K | MATH500 | Avg. |
> | ---------------------- | -------------- | --------------- | -------------- | ----- | ------- | ---- |
> | GPT-4o                 | 63.8           | 36.8            | 50.2           | 94.2  | 74.6    | 63.9 |
> | Mulberry-7B            | 63.1           | 22.8            | 39.6           | 69.1  | 65.2    | 52.0 |
> | Virgo-7B               | 62.3           | 24.0            | 36.7           | 77.4  | 71.4    | 54.4 |
> | MindGYM                | 70.3           | 28.6            | 48.4           | 84.1  | 68.4    | 60.0 |
> | OpenVLThinker-7B       | 70.2           | 25.3            | 47.9           | 78.6  | 67.4    | 57.9 |
> | NoisyRollout           | 72.9           | 28.9            | 52.8           | 79.0  | 68.4    | 60.4 |
> | MM-EUREKA              | 73.0           | 26.9            | 50.3           | 82.5  | 66.8    | 59.9 |
> | Qwen2.5-VL-7B-Instruct | 67.8           | 24.7            | 44.5           | 83.6  | 67.4    | 57.6 |
> | EvolvedGRPO | 74.0           | 30.8            | 51.8           | 85.1  | 73.2    | 63.0 |
>
> > Q5: Mismatch Between Motivation and Method's Effect
>
> A5: Thank you for raising this important question. We will address this from three points:
>
> 1. First, we clarify that EvolvedGRPO is designed to autonomously explore and escalate problem complexity while maintaining logical correctness. It uses the visual perception output as an intermediate state to construct longer CoT. The ultimate goal is to effectively disentangle the complex relationship between vision and text, seamlessly integrating perception into subsequent logical steps.
>
> 2. At the training level, the subsequent augmentation task (e.g., a calculation) imposes stringent precision requirements on the initial perceptual output (z). Any slight inaccuracy in perception is significantly amplified in the final result (D). By managing this process with a carefully designed reward function and filtering mechanism, we leverage this error amplification effect to generate a stronger, more explicit gradient signal. This, in turn, compels the model to disentangle the visual and textual information with much higher precision.
>
> 3. To demonstrate the framework's generality and effectiveness, we leverage Qwen2.5-VL-7B-Instruct to perform complex editing instructions across various real-world vision-language tasks. The results provide strong evidence that our method effectively decouples visual and textual information in diverse scenarios, leading to a significant enhancement of the model's comprehensive reasoning abilities.
>
>    Table 2: Avg performance of all domains on EvolvedGRPO disciplinary reasoning
>
>    | Models                 | Physics | Chemistry | Biology | Avg. |
>    | ---------------------- | ------- | --------- | ------- | ---- |
>    | OpenVLThinker          | 53.8    | 60.6      | 65.0    | 59.8 |
>    | MM-Eureka              | 56.2    | 65.2      | 65.2    | 62.2 |
>    | Qwen2.5-VL-7B-Instruct | 45.4    | 56.4      | 54.0    | 51.9 |
>    | +GRPO                  | 51.6    | 57.1      | 58.6    | 55.8 |
>    | +EvolvedGRPO           | 61.5    | 65.5      | 63.9    | 63.6 |
>    | +EvolvedGRPO w/iter 1  | 51.4    | 56.4      | 57.2    | 55.0 |
>    | +EvolvedGRPO w/iter 2  | 54.2    | 60.8      | 57.6    | 57.5 |
>
>    Table 3: Avg performance of all domains on EvolvedGRPO general medical reasoning
>
>    | Models                 | MedXpertQA-MM | OmniMedVQA | Avg. |
>    | ---------------------- | ------------- | ---------- | ---- |
>    | Qwen2.5-VL-7B-Instruct | 20.3          | 58.4       | 39.4 |
>    | +GRPO                  | 23.8          | 61.0       | 42.4 |
>    | +EvolvedGRPO           | 24.3          | 62.8       | 43.6 |
>
> > Q6: The paper lacks the necessary analysis to prove that this method actually solves the problem it set out to address: reward convergence and training stagnation.
>
> A6: First, we have solved reward convergence by creating a dynamic problem space. As shown in the reward changes provided in the appendix, our adaptive difficulty mechanism ensures that once the model's performance stabilizes on a given problem set, the framework automatically increases the complexity and reasoning chain length. Quantitatively, we further present the gradually changing trends of multiple samples during the enhancement process in the following table:
>
> Table 4: Variables during training
>
> | variables\step | 0    | 30   | 60   | 90   |
> | -------------- | ---- | ---- | ---- | ---- |
> | variance| 0.17 | 0.14 | 0.17 | 0.16 |
> | accuracy       | 0.22 | 0.21 | 0.23 | 0.24 |
>
> As shown in the table, EvolvedGRPO prevents the model from converging to a single optimal policy for a static task distribution, forcing it to constantly adapt and learn.
>
> Second, the practical result of overcoming reward convergence is the alleviation of training stagnation. We provide direct evidence of this by comparing our method against the baseline GRPO framework.
>
>
> Third, as direct evidence against reward convergence, we analyze the reward variance for different valid responses to the same problem. We first randomly sampled a set of N base problems. For each problem, we generated multiple distinct yet valid solutions (e.g., different Chains of Thought) using two methods: (a) for the original problem (baseline), and (b) for the problem after being enhanced by our framework. We then calculated the reward for every solution and computed the variance of these rewards for both sets in the table. We can provide experimental evidence to demonstrate that a complete improvement has been implemented.
>
> > Q7: The paper's image manipulation method (e.g., adding noise or filters) is conceptually very similar to that of NoisyRollout
>
> A7:  Although both our EvolvedGRPO and NoisyRollout edit images, the idea and methodology are different.
>
> In contrast to NoisyRollout, which injects random noise for specific geometric tasks, our method performs semantically meaningful, complex image edits coupled with multi-round text-image interactions, aiming to enhance model capabilities on general-purpose, real-world datasets. (b) Furthermore, in a direct comparison using only image editing, our method outperforms NoisyRollout by an average of 1.0%, highlighting the benefit of our approach. In addition, we have added comparisons with NoisyRollout and our own ablation studies on the lower-level components to better assess the contribution and importance of each part of the system as follows:
>
> Table 5: Extended ablation experiments of EvolvedGRPO
>
> | Models                 | MathVista-mini | MathVision-full | MathVerse-mini | GSM8K | MATH500 | Avg. |
> | ---------------------- | -------------- | --------------- | -------------- | ----- | ------- | ---- |
> | Qwen2.5-VL-7B-Instruct | 67.8           | 24.7            | 44.5           | 83.6  | 67.4    | 57.6 |
> | NoisyRollout           | 72.9           | 28.9            | 52.8           | 79.0  | 68.4    | 60.4 |
> | +GRPO                  | 71.9           | 28.9            | 49.6           | 82.9  | 67.0    | 60.1 |
> | +GRPO & IMAGE EDIT     | 72.1           | 31.8            | 50.9           | 81.3  | 68.8    | 61.0 |
> | +GRPO & TEXT EDIT      | 73.4           | 29.8            | 51.3           | 84.5  | 72.6    | 62.3 |
> | +EvolvedGRPO           | 74.0           | 30.8            | 51.8           | 85.1  | 73.2    | 63.0 |

---

> ### Author Response · Authors · 2025-08-06
> **A kind reminder of the upcoming end of the discussion period**
>
> Dear Reviewer,
>
> I hope this message finds you well. As the discussion period is nearing its end with less than three days remaining, I wanted to ensure we have addressed all your concerns satisfactorily. If there are any additional points or feedback you'd like us to consider, please let us know. Your insights are invaluable to us, and we’re eager to address any remaining issues to improve our work.
>
> Thank you for your time and effort in reviewing our paper.

---

> ### Comment · Reviewer_moty · 2025-08-06
>
> I thank the authors for their detailed rebuttal. The newly provided re-evaluation of baseline models and the comprehensive ablation study have substantially improved the experimental rigor of the paper and successfully addressed some of my initial concerns.
>
> However, I still think that the paper lacks clarity in its methodological description.
>
> Additionally, the central claim that the method solves reward convergence and training stagnation remains unsubstantiated. The evidence provided in the rebuttal is unconvincing, and a promise of future analysis is insufficient at this stage.
>
> Due to these fundamental issues, I will maintain my original rating.

---

> ### Author Response · Authors · 2025-08-09
>
> Thank you for your valuable suggestions. We will address your concerns in two areas: methodological clarification and unsubstantiated claim on solving reward convergence and training stagnation.
>
> **Methodological Clarification:**
>
> In GRPO training for the reward of sampled responses $\\{r\_i\\}^{G}\_{i=1}$, the advantage of each response is:
> $$
> \\hat{A}\_{i,t} = \\frac{r\_i - \\mathrm{mean}\\left(\\{ r\_j \\}\_{j=1}^G \\right)}
> {\\mathrm{std}\\left(\\{ r\_j \\}\_{j=1}^G \\right)}, \\quad \\forall i \\in \\{ 1, \\ldots, G \\}
> $$
> While the advantages are used to calculate the updated gradients:
> $$
> \\mathcal{J}\_{\mathrm{GRPO}}(\\theta) =
> \\mathbb{E} \\left[
>     q \\sim P(Q), \\{ o\_i \\}\_{i=1}^G \\sim \\pi\_{\theta\_{\mathrm{old}}}(O \\mid q)
> \\right] \\\\
> \\frac{1}{G} \\sum\_{i=1}^G \\frac{1}{|o\_i|} \\sum\_{t=1}^{|o\_i|}
> \\left\\{
>     \\min \\left[ r^{i,t}(\\theta) \\hat{A}\_{i,t},
>         \\mathrm{clip} \\left( r^{i,t}(\\theta), 1 - \\epsilon, 1 + \\epsilon \\right) \\hat{A}\_{i,t}
>     \\right]
>     - \\beta D\_{\mathrm{KL}}\\left[ \\pi\_\\theta \\| \\pi\_{\mathrm{ref}} \\right]
> \\right\\}
> $$
> During the multi-round GRPO training with fixed data, the increase in accuracy and the training on fixed problems cause the standard deviation to decrease to zero, leading to reward convergence and, ultimately, training stagnation and collapse. To address this, we introduce EvolvedGRPO, which incorporates adversarial data augmentation to introduce additional information during multi-round training.  We use the following reward to train a question generation model, which generates editing instructions based on external knowledge (eg, math concepts):
> $$
> \\text{Reward} = \\log \\, \\operatorname{sim}(v\_k, v\_0)
> +\\sum\_{i=1}^{k} \\log \\, \\pi\_{\\theta} \\big( a\_i \\mid f\_i(a\_{i-1}) \\big)
> +\\mathbf{format}'
> -\\log \\, \\max \\left( \\pi\_{\\theta}(a\_k \\mid q\_k, v\_k), \\frac{1}{\\mathbf{test}} \\right)
> $$
> We aim to maximize the correctness of step-by-step edits after deconstruction while minimizing the accuracy of answers to synthesized enhancement problems (but ensuring at least one correct answer).

---

> ### Author Response · Authors · 2025-08-09
>
> **Unsubstantiated Claim on Solving Reward Convergence and Training Stagnation**
>
> Next, we will explain, from both theoretical and experimental perspectives, how we have addressed the issue of reward convergence and training stagnation.
>
> 1. Theoretical Analysis
>
> For training data that the model has repeatedly encountered, the accuracy can approach $100\\%$ over multiple learning iterations for part of the data. In such cases, since each evaluation samples only $5$ responses and the reward is variance-normalized, repeated training causes the responses to converge. When all $5$ responses are correct, the variance becomes $0$ and is thus ignored, leading to training stagnation. Furthermore, in order to increase accuracy, the model may omit the reasoning process entirely or produce an incorrect reasoning process that nevertheless yields the correct answer, resulting in negative rewards that can cause training collapse.
>
> In contrast, EvolvedGRPO introduce additional informational complexity during multi-round training, thereby increasing the data's informational entropy and increasing reasoning complexity. This process reduces the model's accuracy, strategically promoting an increase in cognitive load that drives the model to engage in more nuanced multi-step inferential reasoning, preventing training stagnation and fostering further reasoning learning. Furthermore, the integration of additional information and the escalation of difficulty within the synthetic data forces the model to extend its reasoning processes, impeding direct answer memorization and reliance on spurious memorization of erroneous inferential paths and thereby mitigating the risk of training collapse.
>
> 2. Experimental Analysis
>
> **First, we want to demonstrate that we have addressed Reward Convergence.** The practical result of overcoming reward convergence is the alleviation of training stagnation. First, we provide the average variance calculated from the sampling at each corresponding training step, with each training round consisting of 30 steps:
>
> | Varience\Step | 0    | 30   | 60   | 90   |
> | ------------- | ---- | ---- | ---- | ---- |
> | GRPO| 0.17 | 0.10 | 0.08 | 0.06 |
> | EvolvedGRPO   | 0.17 | 0.14 | 0.17 | 0.16 |
>
> We can observe that EvolvedGRPO slows down the decrease in variance, thereby increasing the proportion of effective data and helping to mitigate reward convergence. Additionally, we calculate the average reward for the corresponding intervals:
>
> | Reward\Step | [10,19] | [20,29] | [30,39] | [40,49] | [50,59] | [60,69] | [70,79] | [80,89] |
> | ----------- | ------- | ------- | ------- | ------- | ------- | ------- | ------- | ------- |
> | GRPO| 0.59| 0.60| 0.63| 0.64| 0.66|0.64| 0.66| 0.66|
> | EvolvedGRPO |0.59| 0.60| 0.54| 0.58| 0.60| 0.57| 0.59| 0.62|
>
> We observe that EvolvedGRPO exhibits greater fluctuations during each training round, as it can consistently improve after data augmentation. Specifically, we found that the reward in EvolvedGRPO fluctuates continuously, with the average reward sharply decreasing after each data augmentation before gradually increasing. In contrast, the reward growth in GRPO slows down and eventually stagnates.
>
> The analysis confirms that by generating more complex problems, EvolvedGRPO allows for a richer variety of high-quality solutions, indicating a more diverse and nuanced reward landscape that encourages exploration over exploitation.
>
> **Second, we want to demonstrate that we have addressed training stagnation.** Previously, we compared the accuracy on the validation set and the average inference token length during multi-round training for GRPO and EvolvedGRPO, as shown in Figure 2 of the main text. We observe that after 30 steps, the inference token length gradually stabilizes during the second and third rounds of training, and the score growth on the validation set slows down. This demonstrates that EvolvedGRPO significantly enhances the model's reasoning ability during multi-round training, preventing the training stagnation observed with GRPO.
>
> **Additionally, we have provided an external comprehensive visualization of the entire training process.** During training, GRPO shows little to no improvement after $100$ steps and collapses entirely after approximately $600$ training steps. The response length drops to fewer than $20$ tokens over the next $600$ steps, and the average accuracy decreases by close to $60\\%$, leading to an almost complete loss of reasoning capability.
>
> In contrast to GRPO, our EvolvedGRPO maintains a stable improvement in the model’s reasoning capability during training, with accuracy exhibiting continual fluctuations across iterations and the number of reasoning tokens increasing steadily.

---

### Official Review · Reviewer_nE76 · 2025-06-25

**Clarity:** 3
**Significance:** 2
**Originality:** 2
**Rating:** 4
**Confidence:** 4

**Summary:**

This paper introduces EvolvedGRPO, a reinforcement learning framework aimed at enhancing the reasoning capabilities of Large Vision-Language Models (LVLMs). Built upon the Grouped Relative Policy Optimization (GRPO) method, the key contribution of EvolvedGRPO lies in its progressive instruction evolution mechanism that dynamically generates increasingly difficult multi-modal reasoning tasks. It achieves this by iteratively editing both textual and visual instructions in a self-supervised, adversarial way. Two models—a question generator and an answer model—are jointly trained, where the former adversarially challenges the latter with more difficult tasks. Empirical results show that EvolvedGRPO outperforms other open-source baselines and approaches GPT-4o on several reasoning benchmarks.

**Questions:**

Have you examined whether small perturbations in instructions (e.g., minor edits in visual modality) produce meaningful reward signal changes? Could the reward collapse still persist in late training?
While hallucination reduction is mentioned, how robust is the filtering pipeline?
Can the framework actively regulate difficulty based on performance plateaus? If so, how is this determined?
Could you elaborate on the application potential in real-world settings such as visual question answering for medical or industrial imagery?

**Ethical Concerns:**

["NO or VERY MINOR ethics concerns only"]

**Final Justification:**

I have read the author rebuttal and considered all raised points. I will keep the scores.

**Limitations:**

More detail is needed on how instruction hallucination or adversarial drift is avoided in later stages of evolution.

**Quality:**

3

**Strengths And Weaknesses:**

Strengths: Introduces an effective curriculum-style RL method tailored for LVLMs. Combines adversarial data generation with reward shaping to break reward convergence, a common issue in GRPO.
Weaknesses: The instruction evolution mechanism still relies heavily on implicit assumptions about controllability and reward fidelity. The reward design assumes that small transformations of inputs yield proportionally meaningful reward gradients—this may not always hold in multimodal contexts. The generalization to truly open-ended tasks is unclear.

---

> ### Author Rebuttal · Authors · 2025-07-31
>
> We sincerely appreciate the reviewer's insightful and detailed feedback. We address each of your questions individually below.
>
> > Q1: The instruction evolution mechanism still relies heavily on implicit assumptions about controllability and reward fidelity.
>
> A1: We will focus on analyzing from three aspects: controllability, fidelity, and generalization.
>
> (1.1) Controllability: Our framework enables fine-grained control over reward dynamics by modulating the depth and structure of multimodal reasoning. Increasing the number of inference steps—particularly in long reasoning chains—yields more informative reward variations than shallow perturbations. We further guide the learning process using MLLM-extracted abstract concepts and adversarial editing instructions for text and image editing. The resulting reward trends, which initially decline and then steadily rise across training rounds, reflect the model’s progressive improvement. Appendix D visualizes this evolution, highlighting the controllability of our approach.
>
> (1.2) Fidelity: To ensure data fidelity, we combine a multi-step filtering pipeline with a stepwise Hallucination Estimator to suppress errors and hallucinations during training. In addition, Manual evaluation further confirms the generation of high-quality, semantically valid data throughout the process.
>
> (1.3) Generalization: We conducted additional experiments on several other real-world vision-language tasks, demonstrating that our approach can be effectively adapted to different tasks with minimum modifications.
>
>
> > Q2: Have you examined whether small perturbations in instructions (e.g., minor edits in visual modality) produce meaningful reward signal changes?
>
> A2: We address this concern from two perspectives.
>
> (2.1) Since reward signals in multi-modal reasoning are closely tied to the depth and structure of the reasoning process, changes in entropy at both the visual and textual semantic levels can effectively modulate the reward landscape. Specifically, altering the number of inference steps—particularly in extended reasoning chains involving over 4096 tokens—tends to induce more substantial and informative reward variations than shallow input perturbations. Appendix D visualizes the evolution of reward signals across multiple training rounds, further supporting this observation. For completeness, we plan to integrate these results into the main text in future revisions.
>
> (2.2) We leverage a MLLM to extract abstract concepts and generate adversarial editing instructions, which are then used to guide both text and image editing for multi-modal enhancement. During each round of training, the reward associated with data augmentation initially drops significantly, then gradually increases as training progresses, accompanied by the continuous improvement of the model's capabilities.
>
> > Q3: Could the reward collapse still persist in late training?
>
> A3: Thank you for the concerns about reward collapse. Compared to the original GRPO training, our method not only effectively addresses the issue of reward stagnation caused by repeated training, and also mitigates the reward collapse typically observed after a fixed number of training steps. Specifically, the original GRPO model encounters reward stagnation after approximately 100 steps and collapses entirely after around 600 training steps. In contrast, our EvolvedGRPO framework can maintain stable performance throughout the training process without exhibiting any signs of reward collapse.
>
> > Q4: While hallucination reduction is mentioned, how robust is the filtering pipeline?
>
> A4:  Thank you for the concerns regarding the reduction of hallucinations.
>
> (4.1) Firstly, the filtering pipeline enhances the validation of each step through multi-step disassembly, which can indeed robustly reduce errors. Moreover, the problem generation model incorporates a Hallucination Estimator at each step of reward calculation, ensuring that hallucinations are minimized as much as possible during each training round.
>
> (4.2) To evaluate robustness, we randomly sampled 100 instances and analyzed the distribution of three types of cases across multiple training rounds: (1) valid and effective questions that contribute positively to training; (2) neutral questions that are ineffective but filtered out without impacting training; and (3) erroneous questions that introduce negative effects. The results are as follows:
>
> Table 1: Human evaluation results for hallucination detection
>
> | types  | total | type 1 | type 2 | type 3 |
> | ------ | ----- | ------ | ------ | ------ |
> | number | 100   | 95     | 5      | 0      |
>
> > Q5: Can the framework actively regulate difficulty based on performance plateaus? If so, how is this determined?
>
> A5: The framework can proactively adjust the difficulty. When increasing the difficulty for the question generation model, the reward is computed as the maximum between the actual answer accuracy and `1/test` (where `test` denotes the number of test samples). This guides the difficulty to converge toward `1/test`. As shown in the appendix D, when `1/test = 0.2`, the answer accuracy effectively converges slightly above `0.2`. Additionally, the number of editing steps is increased once the reward stabilizes and the length of the reasoning chain remains unchanged, allowing for further difficulty scaling.
>
> > Q6:  Could you elaborate on the application potential in real-world settings such as visual question answering for medical or industrial imagery?
>
> A6:
>
> We trained on multiple real-world vision-language tasks by replacing mathematical concepts with their corresponding domain-specific counterparts. With the further development of image editing technology, we embed the Flux.Kontext image editing model into the training framework for image editing, to enable richer and more customized image enhancements. The results are as follows:
>
> Table 2: Avg performance of all domains on EvolvedGRPO disciplinary reasoning
>
> | Models                 | Physics | Chemistry | Biology | Avg. |
> | ---------------------- | ------- | --------- | ------- | ---- |
> | OpenVLThinker          | 53.8    | 60.6      | 65.0    | 59.8 |
> | MM-Eureka              | 56.2    | 65.2      | 65.2    | 62.2 |
> | Qwen2.5-VL-7B-Instruct | 45.4    | 56.4      | 54.0    | 51.9 |
> | +GRPO                  | 51.6    | 57.1      | 58.6    | 55.8 |
> | +EvolvedGRPO           | 61.5    | 65.5      | 63.9    | 63.6 |
> | +EvolvedGRPO w/iter 1  | 51.4    | 56.4      | 57.2    | 55.0 |
> | +EvolvedGRPO w/iter 2  | 54.2    | 60.8      | 57.6    | 57.5 |
>
> Table 3: Avg performance of all domains on EvolvedGRPO general medical reasoning
>
> | Models                 | MedXpertQA-MM | OmniMedVQA | Avg. |
> | ---------------------- | ------------- | ---------- | ---- |
> | Qwen2.5-VL-7B-Instruct | 20.3          | 58.4       | 39.4 |
> | +GRPO                  | 23.8          | 61.0       | 42.4 |
> | +EvolvedGRPO           | 24.3          | 62.8       | 43.6 |

---

> > ### Comment · Reviewer_nE76 · 2025-08-05
> >
> > Thanks for the detailed rebuttals, I will keep my scores.

---

> ### Author Response · Authors · 2025-08-07
>
> Thank you for your response and for carefully considering both our rebuttal and the broader reviewer discussion. We fully respect your decision to uphold your original evaluation.
>
> At the same time, we would like to kindly confirm whether all of your concerns have been fully addressed. If there are any remaining issues or clarifications needed, we would be more than happy to provide further responses.
>
> Thank you again for your time and thoughtful review. Wishing you all the best in your future research endeavors.

---

### Official Review · Reviewer_ZeLT · 2025-07-01

**Clarity:** 3
**Significance:** 3
**Originality:** 3
**Rating:** 5
**Confidence:** 2

**Summary:**

One of the main problems with GRPO is that as a model becomes better, its answers become equally as good/accurate. This paper proposes to resolve the 'reward saturation' problem presented by GRPO by presenting a framework that:
(a) Generates progressively more difficult multimodal problems by applying image/text edits using a question generator. The question generator is trained adversarially to propose image/text edits that (i) do not modify the original answer and (ii) are hard for an answer model to solve, but are still solvable.
(b) Utilizes curriculum learning to apply increasing numbers of edits (k) so that the model's reasoning chain lengthens.

The authors test their method on five multimodal reasoning benchmarks (MathVista-mini, MathVision-full, MathVerse-mini, GSM8K, MATH500) and three broad multimodal benchmarks (MMMU, MMStar, AI2D). Their method, which fine-tunes Qwen2.5-VL-7B-Instruct, shows gains over baselines. In ablations, these gains are mostly attributed to the adversarial curriculum.

**Questions:**

1. Address the weaknesses.
2. I really like your method. Have you attempted to use it with real-world vision language tasks?

**Ethical Concerns:**

["NO or VERY MINOR ethics concerns only"]

**Final Justification:**

The authors addressed all my concerns in their rebuttal and I believe that with the new data/experiments, their paper quality is improved.

**Limitations:**

Sort of. In appendix F, there is a brief mention of using more advanced image editing techniques in the future. However, this may not meet this requirement.

**Quality:**

2

**Strengths And Weaknesses:**

Strengths
1. The authors clearly outline the problem that their approach is trying to solve: reward variance collapse and its impact on GRPO training.
2. The authors introduce a novel approach that integrates adversarial instruction generation and curriculum scaling.
3. Their method shows improvement across 8 benchmarks.
4. Ablations that demonstrate complementary effect of both adversarial instruction generation and curriculum scaling.

Weaknesses
1. The authors do not provide detailed information on their image editing pipeline (aside from stating "external tools such as Diffusion models" and similar high level descriptions). It would be advisable to add more details to the appendix.
2. Results are reported from single runs. No standard deviations or statistical significance tests are provided, and confidence intervals are not present. It would be ideal to report results for >= 3 random seeds with confidence intervals and statistical tests against the strongest baseline.
3. They use GPT-4o as a judge and source of reward. This can add biases prevalent in GPT-4o, potentially rewarding responses that look most similar to its own, while penalizing those that are different from its own. The authors may wish to perform a robustness study, using a different model (e.g., Claude) or humans as the judge.
4. Ablations are high-level only. The authors did not isolate lower-level components, such as image/text edits.

---

> ### Author Rebuttal · Authors · 2025-07-31
>
> We sincerely appreciate your constructive and insightful comments! We will explain your concerns point by point.
>
> > **Q1**: More clarification of Image Editing Pipeline.
>
> Thank you for the valuable feedback. In Appendix F, we list external image processing tools such as Prisma Art, OpenCV, and Neural Style. It is evident that the current image editing tools lack the capability to modify complex elements, such as auxiliary lines, hence our focus on modifications related to flipping, cropping, and style transfer. Nevertheless, EvolvedGRPO, being model-agnostic and a self-enhancing framework, can integrate more advanced visual editing techniques such as Flux.Kontext, in a resource-efficient manner.
>
> Thanks to editing techniques like replacement and rotation from Flux.Kontext, we are able to generate varied physical or medical images, enhancing the multimodal difficulty and thereby assisting EvolvedGRPO in achieving better performance improvements. EvolvedGRPO demonstrates strong performance in both complex physical domains (+22.5% in Table 4) and medical domains (+10.7% in Table 5).
>
> > Q2: Robustness of EvolvedGRPO.
>
> A2: As you suggested, we have reproduced the results in three runs for robustness of our experiments. We evaluate the base model Qwen2.5-VL-7B-Instruct, strongest baseline NoisyRollout, and our method EvolvedGRPO with three different seeds. The results are as follows:
>
> Table 1: Multiple evaluations of NoisyRollout and EvolvedGRPO
>
> | Models       | type                    | MathVista-mini | MathVision-full | MathVerse-mini | GSM8K       | MATH500     |
> | ------------ | ----------------------- | -------------- | --------------- | -------------- | ----------- | ----------- |
> | NoisyRollout | mean                    | 72.6           | 28.1            | 52.8           | 79.0        | 68.4        |
> | NoisyRollout | Standard Deviation      | 0.2            | 0.7             | 0.7            | 0.4         | 0.9         |
> | NoisyRollout | 90% Confidence Interval | [72.1,73.1]    | [27.0,29.3]     | [51.6,54.0]    | [78.3,79.7] | [66.8,70.0] |
> | EvolvedGRPO  | mean                    | 73.7           | 30.4            | 52.2           | 85.0        | 73.3        |
> | EvolvedGRPO  | Standard Deviation      | 0.5            | 0.9             | 0.4            | 0.2         | 0.7         |
> | EvolvedGRPO  | 90% Confidence Interval | [73.1,74.3]    | [29.0, 31.9]    | [51.5,52.9]    | [84.5,85.5] | [72.0,74.5] |
>
> > Q3: Analysis of judgement and reward of EvolvedGRPO.
>
> **A3**: We apologize for any confusion regarding the judgement and reward. We would like to address your concerns from two perspectives.
>
> **(3.1)**: Firstly, we would like to clarify that GPT4o is only used to determine whether the answer result is the correct answer. For evaluation, we extended the results of using human evaluation to perform multiple random seed evaluations and calculate the mean. The full results are shown below:
>
> Table 2: Evaluation of EvolvedGRPO via multiple human annotators
>
> | Models      | type | MathVista-mini | MathVision-full | MathVerse-mini | GSM8K | MATH500 |
> | ----------- | ---- | -------------- | --------------- | -------------- | ----- | ------- |
> | EvolvedGRPO | mean | 74.0           | 30.8            | 51.8           | 85.2  | 73.4    |
>
> **(3.2)**: When using reward for mathematical problems, only functional rule-based mathruler are used to calculate higher required rewards, without using LLM to determine whether the answer is correct. In addition, for more complex tasks, rewards can be computed using the base model, thereby avoiding bias preferences in GPT-4o without relying on external models.
>
> > Q4: Ablations are high-level only.
>
> **A4**: Thank you for the suggestion. We have added ablation studies on the lower-level components to better assess the contribution and importance of each part of the system:
>
> Table 3: Extended Ablation Experiments of EvolvedGRPO
>
> | Models                 | MathVista-mini | MathVision-full | MathVerse-mini | GSM8K | MATH500 | Avg. |
> | ---------------------- | -------------- | --------------- | -------------- | ----- | ------- | ---- |
> | Qwen2.5-VL-7B-Instruct | 67.8           | 24.7            | 44.5           | 83.6  | 67.4    | 57.6 |
> | NoisyRollout           | 72.9           | 28.9            | 52.8           | 79.0  | 68.4    | 60.4 |
> | +GRPO                  | 71.9           | 28.9            | 49.6           | 82.9  | 67.0    | 60.1 |
> | +GRPO & IMAGE EDIT     | 72.1           | 31.8            | 50.9           | 81.3  | 68.8    | 61.0 |
> | +GRPO & TEXT EDIT      | 73.4           | 29.8            | 51.3           | 84.5  | 72.6    | 62.3 |
> | +EvolvedGRPO           | 74.0           | 30.8            | 51.8           | 85.1  | 73.2    | 63.0 |
>
> As observed, EvolvedGRPO significantly outperforms most ablation settings across all tasks. With the integration of image and text editing, the capabilities of Qwen2.5-VL in logic-based tasks continue to improve, especially after incorporating text editing. This enhancement predominantly stems from the increased complexity of training data due to cascaded text edits, which in turn strengthen the model's reasoning abilities.
>
> > Q5: Have you attempted to use it with real-world vision language tasks?
>
> **A5**: Thank you for your thoughtful suggestions. Thanks to the scalability of EvolvedGRPO, we have successfully applied Qwen2.5-VL-7B-Instruct to real-world vision-language tasks in the next table. In our testing, EvolvedGRPO has excelled in various fields such as physics, chemistry, and biology, consistently outperforming baseline methods in disciplinary reasoning and general medical QA tasks. This confirms the model’s effectiveness and adaptability in real-world applications.
>
> Table 4: Avg performance of all domains on EvolvedGRPO disciplinary reasoning
>
> | Models                 | Physics | Chemistry | Biology | Avg. |
> | ---------------------- | ------- | --------- | ------- | ---- |
> | OpenVLThinker          | 53.8    | 60.6      | 65.0    | 59.8 |
> | MM-Eureka              | 56.2    | 65.2      | 65.2    | 62.2 |
> | Qwen2.5-VL-7B-Instruct | 45.4    | 56.4      | 54.0    | 51.9 |
> | +GRPO                  | 51.6    | 57.1      | 58.6    | 55.8 |
> | +EvolvedGRPO           | 61.5    | 65.5      | 63.9    | 63.6 |
> | +EvolvedGRPO w/iter 1  | 51.4    | 56.4      | 57.2    | 55.0 |
> | +EvolvedGRPO w/iter 2  | 54.2    | 60.8      | 57.6    | 57.5 |
>
> Table 5: Avg performance of all domains on EvolvedGRPO general medical reasoning
>
> | Models                 | MedXpertQA-MM | OmniMedVQA | Avg. |
> | ---------------------- | ------------- | ---------- | ---- |
> | Qwen2.5-VL-7B-Instruct | 20.3          | 58.4       | 39.4 |
> | +GRPO                  | 23.8          | 61.0       | 42.4 |
> | +EvolvedGRPO           | 24.3          | 62.8       | 43.6 |

---

> > ### Comment · Reviewer_ZeLT · 2025-08-06
> > **Improve my rating**
> >
> > Thank you for taking the time to fully address every concern I had. I will improve my rating to an accept.

---

> ### Author Response · Authors · 2025-08-07
>
> We sincerely appreciate your decision to improve the rating after reviewing our rebuttal. In the revised version, we will incorporate the additional experiments and refine the text accordingly. Thank you once again for your thoughtful review and support.

---

### Official Review · Reviewer_ZaKE · 2025-07-02

**Clarity:** 2
**Significance:** 2
**Originality:** 2
**Rating:** 4
**Confidence:** 3

**Summary:**

The paper introduces EvolvedGRPO, a new framework designed to enhance the reasoning capabilities of VLM. The authors identify a key problem with existing reinforcement learning methods like GRPO: as LVMs improve, the reward scores for different answers to the same multi-modal question tend to converge. This lack of reward variance hinders effective learning. To solve this, EvolvedGRPO progressively generates more complex problems that are matched to the model's current abilities through an adversarial process involving a "question generation model" (Q) and an "answer model" (A).  The two models are trained adversarially, with Q rewarded for making challenging yet solvable problems and A rewarded for correct answers, while a curriculum learning strategy gradually increases the number of editing instructions to guide the model toward more complex reasoning.

**Questions:**

See Weaknesses.

**Ethical Concerns:**

["NO or VERY MINOR ethics concerns only"]

**Final Justification:**

The author has addressed my main issues.

**Limitations:**

Yes

**Quality:**

2

**Strengths And Weaknesses:**

**Strengths**

- The paper is well-written and easy to follow.
- The paper identifies and addresses a specific, critical bottleneck in training LVMs.  The proposed EvolvedGRPO framework is an innovative solution that tackles this by progressively increasing the difficulty of training data to match the model's growing capabilities.
- The authors demonstrate its EvolvedGRPO effectiveness across a wide range of benchmarks.

**Weaknesses**
- Limited Scope of Demonstrated Instruction Evolution: The text editing instructions are primarily described as performing "mathematical transformations" to increase reasoning steps. This approach is well-suited for the mathematical and logical reasoning benchmarks tested, but its applicability to more abstract, qualitative, or common-sense reasoning tasks is less clear.
- Concern over Quality Control and Cumulative Error: The framework's success is highly dependent on the quality and fidelity of the instructions produced by the question generation model (Q). Because the method increases problem difficulty through multi-step iteration , small errors or logical deviations introduced in early steps can be amplified in subsequent ones, potentially creating samples that are complex in appearance but logically flawed or unfaithful to the original intent. While the paper mentions using a judge model and rule-based checkers for filtering, it does not detail how robust this filtering mechanism is against subtle, accumulating errors that might not violate a simple rule but still corrupt the problem.

---

> ### Author Rebuttal · Authors · 2025-07-31
>
> We sincerely thank you for the valuable comments! We are encouraged to see that our work can enhance subsequent research endeavors. We will address your concerns point by point.
>
> > **Q1**: Limited Scope of Demonstrated Instruction Evolution.
>
> A1: Thank you for raising the important concern regarding the scope of demonstrated instruction evolution in our manuscript. We first leverage Qwen2.5-VL-7B-Instruct to perform more abstract (e.g., conceptual reasoning) and commonsense (e.g., real-world knowledge)-based editing instructions, and then showcase the universal adaptability and scalability of our evolved instruction generation process.
>
> (1.1) Evaluation on More Real-World Vision-Language Tasks Using Qwen2.5-VL
>
> We conduct instructional evolution on various disciplinary reasoning and general medical QA tasks with Qwen2.5-VL-7B-Instruct.
>
> The results are presented in Tables 1 and 2. Using Qwen2.5-VL-7B-Instruct as the backbone, EvolvedGRPO continues to demonstrate its effectiveness across multiple real-world domains such as physics, chemistry, and biology, outperforming all baseline methods on both disciplinary reasoning and general medical QA scopes. This confirms that EvolvedGRPO is model-agnostic and effective across various types of real-world vision-language tasks.
>
> Table 1: Avg performance of all domains on EvolvedGRPO disciplinary reasoning
>
> | Models                 | Physics | Chemistry | Biology | Avg. |
> | ---------------------- | ------- | --------- | ------- | ---- |
> | OpenVLThinker| 53.8    | 60.6      | 65.0    | 59.8 |
> | MM-Eureka | 56.2    | 65.2      | 65.2    | 62.2 |
> | Qwen2.5-VL-7B-Instruct | 45.4  | 56.4 | 54.0 | 51.9 |
> | +GRPO    | 51.6    | 57.2 | 58.6| 55.8 |
> | +EvolvedGRPO  | 61.4   | 65.4 | 64.0 | 63.6 |
> | +EvolvedGRPO w/iter 1 | 51.4 | 56.4 | 57.2 | 55.0 |
> | +EvolvedGRPO w/iter 2 | 54.2 | 60.8      | 57.6 | 57.5 |
>
> Table 2: Avg performance of all domains on EvolvedGRPO general medical reasoning
>
> | Models                 | MedXpertQA-MM | OmniMedVQA | Avg. |
> | ---------------------- | ------------- | ---------- | ---- |
> | Qwen2.5-VL-7B-Instruct | 20.3          | 58.4       | 39.4 |
> | +GRPO                  | 23.8          | 61.0       | 42.4 |
> | +EvolvedGRPO           | 24.3          | 62.8       | 43.6 |
>
> (1.2) Analysis on the scalability of EvolvedGRPO
>
> EvolvedGRPO has demonstrated robust performance across multiple domains by employing a single iteration in its operations. As suggested, we have conducted additional experiments with more iterations to examine its scalability. The outcomes showcase the universally adaptable and scalable nature of our evolved instruction generation process, as detailed in the last two rows of Table 1.
>
>
> > **Q2**: Concern over Quality Control and Cumulative Error.
>
> A2: Thank you for your questions. We will address them from three points:
>
> (2.1) In our process reward function, we explicitly penalize large discrepancies between step t and step 0 through model-inferred output discrepancies instead of naive rule-based methods. In this way, we adaptively reduce the cumulative error introduced at each step and mitigate the overall error accumulation in multi-step enhancement, even in more complex physical (+22.5% in Table 1) and medical domains (+10.7% in Table 2).
>
> (2.2) To better illustrate the effectiveness of quality control, we present detailed visualizations of reward dynamics during the multi-round training process in Appendix D. During each round of training, the reward associated with data augmentation initially drops significantly, then gradually increases as training progresses.
>
> (2.3) To better illustrate the reduction of Cumulative Error, we quantitatively analyze the accumulation of errors over time t during the progressive enhancement of batch samples. We measure this by the inverse of the similarity between step t and step 0 using human evaluation. The results are presented in Table 3.
>
> Table 3: Cumulative error during training
>
> | step         | 0    | 30   | 60   | 90   |
> | ------------ | ---- | ---- | ---- | ---- |
> | Accumulation | 1.0  | 1.06 | 1.08 | 1.08 |

---

> > ### Comment · Reviewer_ZaKE · 2025-08-06
> >
> > Thank you for the clarifications and for addressing my concerns. I'd like to raise my score to 4.

---

> > > ### Author Response · Authors · 2025-08-07
> > >
> > > Thank you very much for reviewing our rebuttal and for raising the score. We sincerely appreciate your valuable feedback and support. In the revised version, we will carefully incorporate both the theoretical analysis and experimental results as suggested.

---

### Comment · Area_Chair_MjkC · 2025-08-05

Dear Reviewers, given the authors' response, if you have not done so, please raise any remaining questions and/or concerns in a timely fashion so the authors have a chance to reply.

I remind you that Reviewers must participate in discussions with authors before submitting "Mandatory Acknowledgement”.

Thank you for your work.

---

### Author Response · Authors · 2025-08-09
**General Response**

Dear Reviewers, ACs, and SACs:

We are truly grateful for the time and thoughtful feedback you have provided, which have played a crucial role in strengthening our manuscript! Overall, we are encouraged by your finding that:

- The paper introduces an effective curriculum-style RL method tailored for LVLMs. It combines adversarial data generation with reward shaping to **break reward convergence, which solves an urgent issue in multi-modal post training**. (Reviewer nE76)
- The paper clearly outlines the challenge problem and introduces a novel approach.  The method improves performance on 8 benchmarks, with ablations showing **complementary effects of adversarial instruction generation and curriculum scaling**. (Reviewer ZeLT)
- The paper is well-written and accessible, addressing **a critical bottleneck in training LVMs by progressively increasing the difficulty of the training data** to align with the model's evolving capabilities. It also demonstrates the approach's effectiveness across a wide range of benchmarks. (Reviewer ZaKE)
- The paper achieves **excellent results on several challenging multi-modal reasoning** benchmarks, and the paper's motivation is compelling. (Reviewer moty)

To comprehensively address the reviewers’ concerns, we have conducted several additional experiments:

- Extending evaluation of the baseline and our method via multiple tests using diverse metrics, with mean, standard deviation, and confidence intervals to ensure robustness.
- Experimenting on more Real-World Vision-Language Tasks using Qwen2.5-VL, including multidisciplinary reasoning and general medical reasoning, confirming the model’s effectiveness and adaptability in real-world applications.
- Evaluating the robustness of the model in data generation and filtering by conducting human assessments that categorized the data into various types to determine its quality.
- Quantitatively analyzing the accumulation of errors over time during the progressive enhancement of batch samples, further exhibiting strong error control capability.
- Extending our ablation studies on the lower-level components to better assess the contribution and importance of each part of the system.
- Compared our method with the baseline GRPO, providing direct evidence that EvolvedGRPO maintains robust learning at the same steps where reward stagnation and collapse occur.

We have also clarified the following key points:

- Clarifying a self-evolution mechanism that operates entirely through internal rewards.
- Analyzing how to improve data quality through model reinforcement learning training and data filtering.
- Presenting a theoretical discussion on how to increase reward variance and mitigate reward stagnation.
- Providing a detailed explanation of the specific procedures for image and text editing, analyzing the importance of each module, and extending the approach to new datasets.

Furthermore, we have included detailed responses to the additional questions posed by Reviewer moty.

These experiments and clarifications will be incorporated into either the main body or the appendix of our paper. We sincerely thank all reviewers once again for the valuable suggestions! Thank you!

*Best regards,*

*NeurIPS 2025 Conference Submission95 Authors*

---

### Note · Authors · 2025-08-16

Dear Reviewers, ACs, and SACs:

We are truly grateful for the time and thoughtful feedback you have provided. Overall, we are encouraged by your finding that:

1. The paper introduces an effective curriculum-style RL method tailored for LVMs to **break reward convergence, which solves an urgent issue in multi-modal post training**. (Reviewer nE76)
2. The paper clearly outlines the challenging problem and shows **complementary effects of adversarial instruction generation and curriculum scaling**. (Reviewer ZeLT)
3. The paper is well-written and accessible, addressing **a critical bottleneck in training LVMs by progressively increasing the difficulty of the training data**. (Reviewer ZaKE)
4. The paper achieves **excellent results on several challenging multi-modal reasoning benchmarks**, and the paper's motivation is compelling. (Reviewer moty)

To comprehensively address the reviewers’ concerns, we have conducted several additional experiments and analyses:

1. We provide **theoretical and empirical discussions** to demonstrate how EvolvedGRPO **increases reward variance and mitigates reward stagnation**. Furthermore, we extend the evaluation with **mean, standard deviation, and confidence intervals** to show its robustness.
2. We highlight the advantages of EvolvedGRPO with quantitative analysis, evaluating the robustness of EvolvedGRPO through human assessment, showing that it adaptively adjusts sample complexity and conducts self-checking data filtering to **ensure data quality and prevent error accumulation**.
3. We extend EvolvedGRPO to **multidisciplinary reasoning and general medical reasoning** to show its effectiveness and adaptability in real-world applications.

We are trying to polish our paper in the revised version for the community. Specifically:

1. We will revise the methodological descriptions to clarify the novelty of EvolvedGRPO in mitigating reward convergence and training stagnation.
2. We will provide more detailed image and text editing procedures in the main body for improved clarity.
3. We will add theoretical and experimental discussions to further demonstrate the advantages of our approach.
4. We will incorporate more visualizations (e.g., error dynamics and variance trends) to illustrate stability.

These experiments and clarifications will be incorporated into the main body for clarity. We sincerely thank all reviewers once again for the valuable suggestions!

*Best regards,*

*NeurIPS 2025 Conference Submission95 Authors*

---

### Decision · Program_Chairs · 2025-09-17

**Decision:**

Accept (poster)

**Comment:**

Advanced large vision-language models require new efficient  reinforcement learning techniques in particular for multimodal reasoning. Approaches such as  Grouped Relative Policy Optimization (GRPO) assign varying reward scores to different responses from the same prompt to estimate gradients. However, designing such rewards is very challenging in multimodal contexts.  In this paper, the authors introduce  EvolvedGRPO, a framework  that iteratively updates the difficulty of samples progressively using editing with a generation model that  generates multi-modal data editing instructions (texts and images) to augment the dataset.

According to most reviewers comments, the paper is well-written, and addresses a challenging  training issue of LVMs. The proposed approach is also tested using a large variety of experiments. During rebuttal, the authors addressed most of the reviewers concerns on the methodology and implementation details and conducted additional experiments to support the robustness of the approach. They tested the instructional evolution on various disciplinary reasoning and general medical QA tasks with Qwen2.5-VL-7B-Instruct. They also added ablation studies on the lower-level components of the model, the approach consistently outperforming the ablation settings.
Overall, the rebuttal clarified the procedure and the authors added various numerical results which support the performance of the proposed framework in a large variety of settings. A revised version including all comments would be valuable to the ML community, (details on the text and images editing to clarify the algorithm for a larger audience) . The authors are encouraged to clarify the remaining concerns  about reward convergence and training stagnation (at least discuss its long-term viability follwing the discussion with reviewer moty).